# The push and pull of abandoned channels: How floodplain processes and healing affect avulsion dynamics and alluvial landscape evolution in foreland basins

Harrison K. Martin[1], Douglas A. Edmonds[1]

[1]Department of Earth and Atmospheric Sciences, Indiana University, Bloomington, Indiana, 47408, United States of America

*Correspondence to*: Harrison K. Martin (hkmartin@iu.edu)

**Abstract**

River avulsions are an important mechanism by which sediment is routed and emplaced in foreland basins. However, because avulsions occur infrequently, we lack observational data that might inform where, when, and why avulsions occur and these issues are instead often investigated by rule-based numerical models. These models have historically simplified or neglected the effects of abandoned channels on avulsion dynamics, even though fluvial megafans in foreland basins are characteristically covered in abandoned channels. Here, we investigate the pervasiveness of abandoned channels on modern fluvial megafan surfaces. Then, we present a physically based cellular model that parameterizes interactions between a single avulsing river and abandoned channels in a foreland basin setting. We investigate how abandoned channels affect avulsion set-up, pathfinding, and landscape evolution. We demonstrate and discuss how the processes of abandoned channel inheritance and transient knickpoint propagation post-avulsion serve to shortcut the time necessary to set-up successive avulsions. Then, we address the idea that abandoned channels can both repel and attract future pathfinding flows under different conditions. By measuring the distance between the mountain-front and each avulsion over long ($10^6$ to $10^7$ years) timescales, we show that increasing abandoned channel repulsion serves to push avulsions farther from the mountain-front, while increasing attraction pulls avulsions proximally. Abandoned channels do not persist forever, and we test possible channel healing scenarios (deposition-only, erosion-only, and far-field directed) and show that only the final scenario achieves dynamic equilibrium without completely filling accommodation space. We also observe megafan growth occurring via ~100,000 year cycles of lobe switching, but only in our runs that employ deposition-only or erosion-only healing modes. Finally, we highlight opportunities for future field work and remote sensing efforts to inform our understanding of the role that floodplain topography, including abandoned channels, plays on avulsion dynamics.

## 1. Introduction

Avulsions, the wholesale relocations of rivers into new positions on their floodplains, are a primary control on how water and sediment are routed through alluvial landscapes (Mackey and Bridge, 1995). The predominant conceptual model presents avulsions as requiring two necessary components: a set-up, and a triggering event that causes bank failure and avulsion (Slingerland and Smith, 2004). However, there is a lack of observational data on each of these necessary components because avulsions occur infrequently (Edmonds et al., 2016). Instead, avulsion dynamics are often explored using concept-driven numerical models. One such form is cellular models, which seek to reduce the system to the components necessary to reproduce a natural phenomenon (Jerolmack and Paola, 2007). For planform avulsion models, this usually entails some description of sediment transport and deposition along an active channel and associated floodplain, and semi-heuristic rules for how avulsions are set-up, initiate, and pathfind (Hajek and Wolinsky, 2012). However, models have historically simplified or neglected the effect of abandoned channels on avulsion dynamics (Pelletier et al., 2005; Reitz et al., 2010). The relict topographic highs and lows associated with alluvial ridges, levees, and abandoned channels should affect both avulsion set-up and avulsion pathfinding (Leeder, 1977; Allen, 1978; Jerolmack and Paola, 2007; Reitz et al., 2010). These effects could manifest as repulsion, if an approaching avulsing flow is restricted from entering an abandoned channel because of the topographic high formed by remnant levees, or attraction, if flow is routed along the topographic lows of former channel pathways (Edmonds et al., 2016).

The large, fan-shaped, low-relief fluvial megafans that exist where rivers leave lateral confinement and enter foreland basins are ideal locations to study the interaction between avulsions and abandoned channels (Fig. 1A; Leier et al., 2005; Weissmann et al., 2010). Fluvial megafans have some of the highest avulsion rates in the observational record (Valenza et al., 2020) and, in contrast to deltaic fans, have been qualitatively described as hosting abundant abandoned channels (e.g., Assine and Soares, 2004; Rossetti and Valeriano, 2007; Chakraborty et al., 2010; Bernal et al., 2011; Weissmann et al., 2013). However, we lack a detailed evaluation of the prevalence or distribution of this channelization, which is important to understand the degree to which avulsions may interact with abandoned channels.

In this paper, we present observational data on the channelization of fluvial megafan surfaces in alluvial foreland basin settings and we use these observations to motivate a physically based numerical model that parameterizes interactions between an avulsing river and abandoned channels in a subsiding basin. Our model implements tuneable abandoned channel dynamics that influence how abandoned channels affect pathfinding and are removed from the floodplain. Incorporating abandoned channel floodplain dynamics allows us to assess how abandoned channels affect where, when, and why avulsions occur. We demonstrate that abandoned channels, their interactions with future pathfinding flows, and the way they are removed from the floodplain are all important controls on avulsion locations, dynamics, and resulting foreland basin deposition and geomorphology that should be considered in future models and studies.

## 2. Background information

If abandoned channels are common features on floodplain surfaces, it is reasonable to expect that they should affect how avulsions find new pathways. Despite this, most previous models have assumed abandoned channels have no effect on future avulsion pathfinding (Leeder, 1977; Ratliff et al., 2018), or act as universal repulsors (Allen, 1978; Bridge and Leeder, 1979) or universal attractors (Sun et al., 2002; Jerolmack and Paola, 2007; Reitz et al., 2010). The reality seems to be somewhere in-between these endmembers. Both remote sensing (e.g. Edmonds et al., 2016; Valenza et al., 2020) and stratigraphic (e.g. Mohrig et al., 2000; Chamberlin and Hajek 2015, 2019) evidence suggests that avulsions commonly reoccupy former abandoned channel pathways. However, if abandoned channels retain the superelevation that caused avulsion and abandonment, then that superelevation would topographically repel later pathfinding events (Leeder, 1977; Allen, 1978).

The earliest alluvial stratigraphy models to connect avulsions to alluvial architecture encoded the effects of abandoned channels on avulsion pathfinding differently. The pioneering Leeder (1977), Allen (1978), and Bridge and Leeder (1979) models created 2D vertical slices of stratigraphy resulting from channel avulsion across a basin over time. These models required heuristic rules about where successive rivers would be emplaced, including choosing locations randomly (Leeder, 1977), according to lowest elevation, (Bridge and Leeder, 1979), or randomly with additional elements of local abandoned channel repulsion (Allen, 1978). While their resulting stratigraphic sections were fairly insensitive to these differences (Hajek and Wolinsky, 2012), modern successors of these cross-section alluvial architecture models (e.g., Chamberlin and Hajek, 2015, 2019) have demonstrated that choosing different avulsion emplacement rules exerts a significant control on resulting stratigraphy. These rules position future channels along random (along a uniform distribution), compensational (at the lowest topographic elevation), or clustered (likelier to be nearer to the previous channel position) distributions. While these rules prescribe the cross-basin location of successive channels without needing to resolve planform pathfinding, and the compensation rule contains an essence of repulsion as successive elevated abandoned alluvial ridges are left behind, it is unclear how flow routing due to abandoned channel repulsion or attraction further upstream affects or reflects each rule.

To avoid making explicit assumptions about channel emplacement, a parallel lineage of avulsion models resolve avulsion dynamics and pathfinding in a planform basin. These avulsion models necessarily require rules for hydrodynamics, sediment transport, and avulsion set-up, initiation, pathfinding, and stabilization, but allow for a more sophisticated interaction between avulsion pathfinding and floodplain topography (including abandoned channels) than can be resolved in cross-section models (Hajek and Wolinsky, 2012). Whenever these models incorporate topographic steepness (with or without random noise) into avulsion pathfinding, and do not instantly erase the topographic alterations made by abandoned channels on landscapes, pathfinding is controlled by abandoned channels (e.g., Mackey and Bridge, 1995; Coulthard et al., 2002; Sun et al., 2002; Jerolmack and Paola, 2007; Reitz et al., 2010). In addition to affecting avulsion dynamics, the rate at which these abandoned channels (and associated alluvial ridges and scours) are removed should affect avulsion pathfinding and hence landscape evolution. There are not many observations of abandoned channel healing rates, and those that exist are generally limited to sedimentation rates in oxbow lakes hydraulically connected to active channels (e.g. Cooper and McHenry 1989;

Rowland et al., 2005; Wren et al., 2008; Kołaczek et al., 2017). As such, it is unclear in models whether to treat abandoned channels as healing instantly, persisting indefinitely, or some intermediary. As a broad classification, there are at least four assumptions that can describe the fate of these abandoned channels: i) avulsed channels do not leave behind abandoned channels on floodplains (instant healing; Ratliff et al., 2018, 2021), ii) abandoned channels do not change after avulsion (no healing), iii) abandoned channels are instantly healed after some fixed number of timesteps (Reitz et al., 2010), or iv) abandoned channels are healed gradually over time by adjusting their channel-base and/or levee-top elevations (Jerolmack and Paola, 2007). The first three assumptions do not allow abandoned channels to act as both repulsors and attractors, which is inconsistent with observations of avulsing rivers (Edmonds et al., 2016; Valenza et al., 2020). Further, the first assumption generates no abandoned channel topography on floodplains whatsoever.

Additionally, if one assumes that abandoned channels do heal, the mode of abandoned channel healing is unclear. While little is known about the constructive and destructive processes in action on floodplains, we can speculate on the evolution of abandoned channels using observations from both degradational and aggradational settings (Hartley et al., 2010b). During floods, overbank sediments can preferentially deposit in abandoned channel topographic lows (Wolman and Eiler, 1958; Schmudde, 1963; Bridge and Leeder, 1979; Lewis and Lewin, 1983; Farrell, 1987; Nanson and Croke, 1992; Tooth et al., 2002; Jerolmack and Paola, 2007; Toonen et al., 2012). This 'bottom-up' healing, however, can be undone in some cases by scouring from future flooding events (Wolman and Leopold, 1957; Wolman and Eiler, 1958; Schmudde, 1963; Bridge and Leeder, 1979). The relative degree of levee erosion or deflation (something we call 'top-down' healing) is unknown. Top-down healing is plausible if a combination of diffusive sediment transport, weathering, and fluvial erosion during floods erode or diffuse topographic highs on the floodplain (Hack and Goodlett, 1960; Burkham, 1972; Zwoliński, 1992; Gabet, 2000; Croke et al., 2013). High elevation on floodplains could be eroded during subsequent flood events, or could conceivably be gradually diffused; while it is not clear whether diffusion should also describe the evolution of alluvial floodplain topographic highs, biologic disturbance is often high (Richards et al., 2002; Steiger et al., 2005). Complicating matters, sediment deposition during overbank flows has also been observed atop flat or even positive floodplain topography, promoting self-sustaining topography that also hinders abandoned channel healing (Jahns, 1947; Wolman and Eiler, 1958; Schmudde, 1963; Nanson, 1980).

In summary, there are unanswered questions about the fates and rates of abandoned channel floodplain topography. There are virtually no data that describe how these landforms change through time once they are abandoned on the floodplain. This is an important knowledge gap because the primary mode of sediment transport and emplacement in this depositional environment is via alluvium deposited by and between avulsions. It is conceivable that the gross morphology of foreland basins and their deposits depends on the interaction between avulsing rivers and abandoned channels.

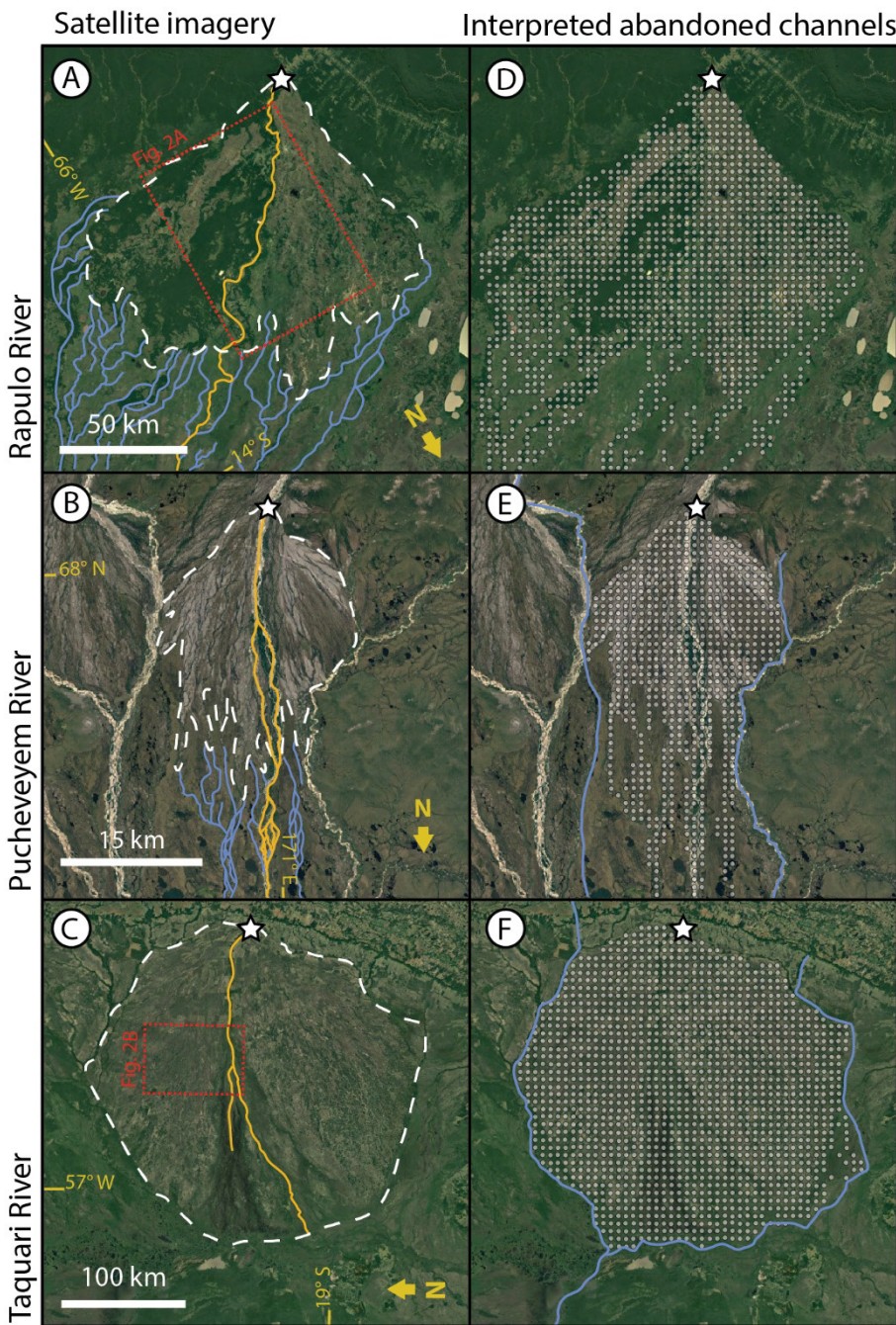

Figure 1: (a-c) Remote sensing images and (d-f) abandoned channel maps for three fluvial megafans. The fans are located along the Rapulo River in Bolivia (a,d), the Pucheveyem River in Russia (b,e), and the Taquari River in Brazil (c,f). Note the downstream transition between distributive, densely channelized abandoned channel networks to tributive, sparsely channelized networks. Dashed white lines in (a-c) are interpreted megafan boundaries (see text for details), and white stars mark megafan apices. Blue lines in (a-c) show interpreted abandoned channel pathways outside of the megafan boundary. Solid gold lines show active channels. All satellite images are USGS/NASA Landsat/Copernicus, © Google Earth.

## 3. Observations of modern fluvial megafan surfaces

In order to motivate considering the importance of abandoned channels on avulsion dynamics, we must investigate the pervasiveness of abandoned channels in landscapes where avulsions are common. To do this, we created maps of abandoned channels on a non-exhaustive set of three megafans (Fig. 1) that represents a range of megafan sizes and settings with typical appearances (Hartley et al., 2010a; Weissmann et al., 2010). This set includes the well-studied Taquari megafan (e.g., Assine, 2005; Makaske et al., 2012; Zani et al., 2012). Following previous work (Rossetti and Valeriano, 2007; Bernal et al., 2011), we combine Google Earth, Landsat visual imagery, and bare-earth topography to identify abandoned channels on these megafans. For elevation data, we use the BEST (Bare-Earth Srtm Terrain) elevation model, which uses vegetation maps and satellite lidar to reveal bare-earth topography by correcting for vegetation elevations present in radar-derived topography (O'Loughlin et al., 2016; Moudrý et al., 2018). On top of each megafan, we overlaid a rasterized grid with square cells with dimensions that corresponded to roughly five channel widths, similar to the resolution of the cellular model that is described later. Within each cell we marked whether there was topographic or visual evidence of abandoned channels (Fig. 1D-F). Evidence of abandoned channels consisted of identified channelized features with long axes generally oriented toward the apex of the fan, with widths approximately equal to the active channels on the fan. Abandoned channels were usually visible in satellite imagery, but in areas with dense tree canopies, we looked for channel-like pathways delineated by differences in coloration against the adjacent floodplain. These differences result from the historical presence of an active channel (Bernal et al., 2011); abandoned channels may have coloration that is lighter (due to sediment emplacement; Valenza et al., 2020) or darker (due to increased vegetative density associated with additional standing or groundwater). Where possible, we used the topographic data in tandem with the visual data to confirm that a cell contained an abandoned channel.

### 3.1. Remote sensing results:

These three fluvial megafans have abundant abandoned channels within their boundaries (Fig. 1). Megafan boundaries were drawn to encapsulate regions of positive relief and greater slope relative to the surrounding basin (dashed white lines, Fig. 1A-C). Within these boundaries, between 95% (Rapulo) and >99% (Pucheveyem and Taquari) of cells on megafan surfaces contained interpreted abandoned channel features. Downstream of the megafan boundary there is a transition from distributive to tributive planform morphologies (Fig. 1A,B), wherever they are not bounded by topography or an axial river (Fig. 1C; cf. methodology of Hartley et al., 2010a). In contrast to other distributary fan systems like alluvial fans or some deltas, within the fan we usually observed a single active channel with one or multiple threads and occasional bifurcation (Hartley et al., 2010a). This suggests that, rather than hosting many contemporaneous distributary channels, the distributive nature of megafans arises over time through repeated avulsions along a small number of active channels (Weissmann et al., 2010).

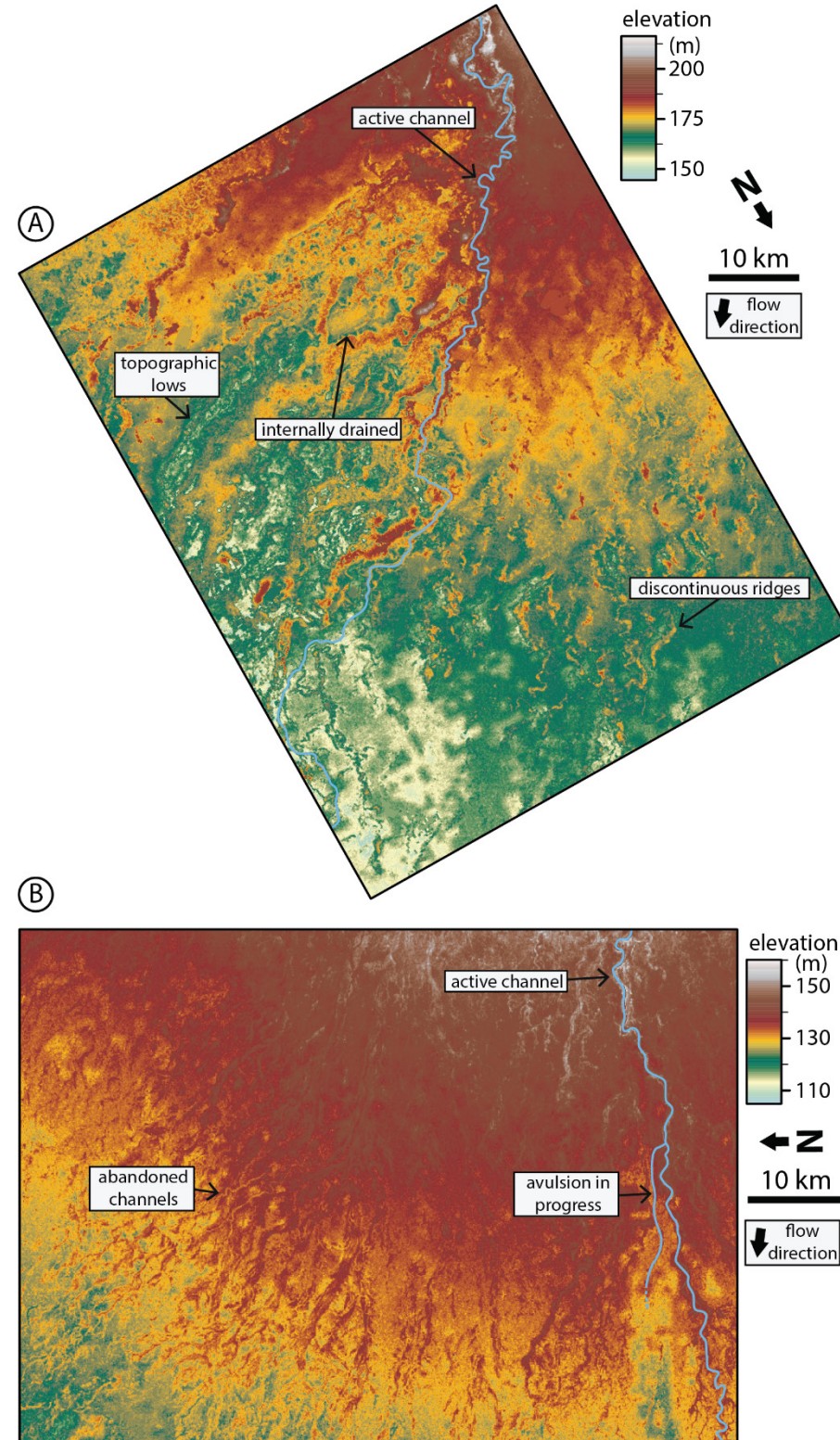

157

**Figure 2: Bare-earth digital elevation models (O'Loughlin et al., 2016) of floodplain topography on the Rapulo (a) and Taquari (b) megafans. Locations are in Fig. 1. Floodplains are densely channelized by abandoned channels with visible topographic highs and lows corresponding to levees or alluvial ridges and channel beds, respectively. Note the presence of discontinuous alluvial ridges. Also note that the colorbar is not perceptually uniform, meaning that small changes in certain elevations ranges are highlighted more drastically than others; this is done to emphasize low-relief features relative to the overall fan slope.**

We observed that abandoned channels can be both topographic highs (associated with levees or alluvial ridges) and topographic lows (associated with abandoned channels that have not been fully in-filled with sediment) relative to surrounding floodplains (Fig. 2). In some portions of the fan there were 'internally drained' areas surrounded by topographic abandoned channel highs (Fig. 2A). These abandoned alluvial ridges were not necessarily spatially continuous in the downstream direction, often forming discontinuous ridges (Fig. 2; Rossetti and Valeriano, 2007). The topographic data were collected during an ongoing avulsion on the Taquari fan, and the avulsion location is immediately adjacent to a topographic low on its floodplain (Buehler et al., 2011). Multiple avulsions on the Rapulo megafan during the Landsat observation period have also initiated into local topographic lows adjacent to the channel (Edmonds et al., 2022).

**3.2. Megafan floodplain topography discussion:**

The degree of floodplain channelization observed and interpreted on megafan surfaces in Fig. 1 and Fig. 2 compares well to the model results of Jerolmack and Paola (2007) and Reitz et al. (2010), and suggests that most avulsions should interact with abandoned channels. Following this, we envision at least two aspects of avulsion dynamics that can be influenced by the presence of abandoned channels and can be easily incorporated into a model.

**3.2.1 Avulsion set-up & initiation**

The most common conception of avulsion set-up is superelevation, whereby in-channel deposition outpaces deposition in the surrounding floodplain, leading to a perched channel that transports water and sediment less efficiently than some novel path on the floodplain (Bryant et al., 1995; Slingerland and Smith, 2004). On a flat, featureless floodplain where subsidence is uniform along-strike, the time to achieve superelevation ($T_A$, years) for some arbitrary point along a river is commonly (e.g., Jerolmack and Mohrig, 2007; Jerolmack, 2009; Martin et al., 2009; Reitz et al., 2010; Moodie et al., 2019) approximated as

$$T_A = \frac{\beta * h_{chan}}{A_{chan} - A_{fp,tot}} \qquad (1)$$

where $\beta$ is a non-dimensional channel depth fraction (generally assumed to be 0.5-1.0; Mohrig et al., 2000), $h_{chan}$ is the channel depth (meters) at a particular point in the river, $A_{chan}$ is the in-channel-bed aggradation rate (meters per year), $A_{fp,tot}$ is the total floodplain aggradation rate (meters per year). Conceptually, this superelevation timescale is equal to the time necessary for the channel bed to aggrade some specified fraction of a channel depth (Fig. 3A).

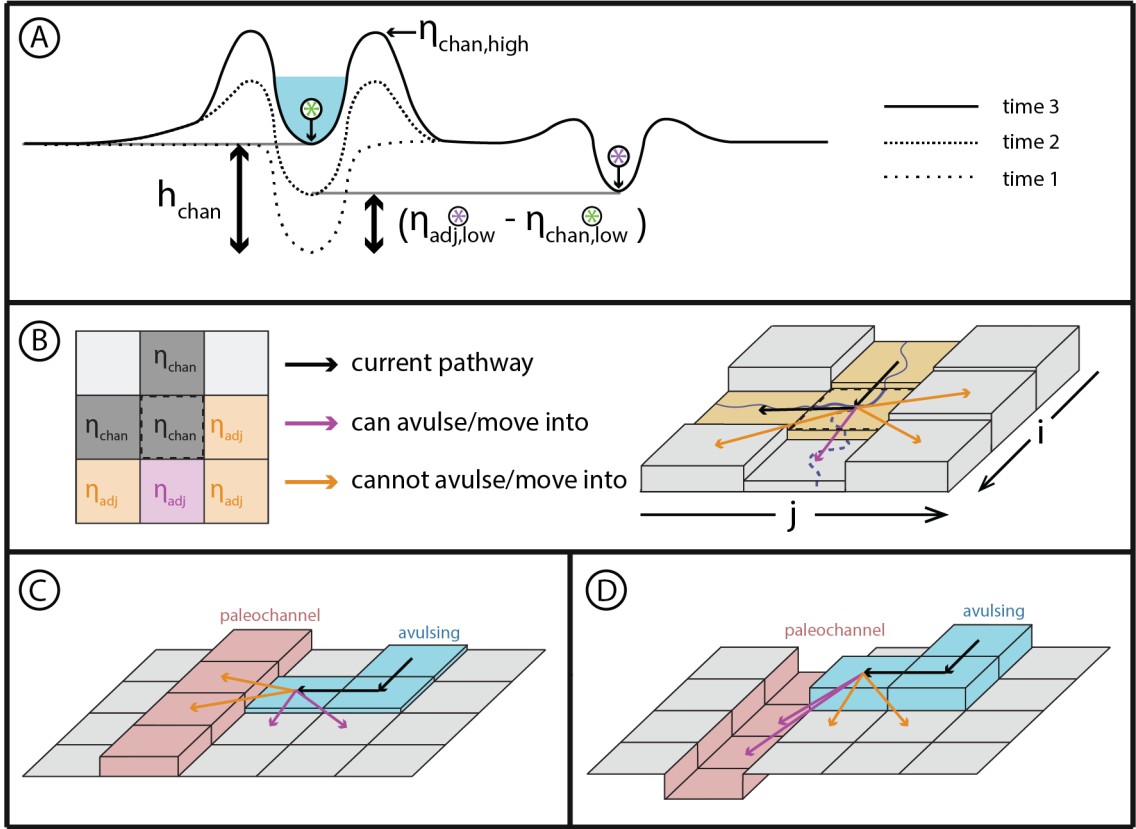

188

**Figure 3: (a)** Conceptual understanding of avulsion set-up by superelevation, and how the magnitude of in-channel aggradation necessary to achieve aggradation (black bidirectional arrows) differs if abandoned channels are ignored (left of active channel; Eq. (1)) or considered (right of active channel; Eq. (2)). **(b)** Representation of avulsion set-up by superelevation in a 2D cellular setting that considers adjacent elevations. Cells in the left panel are marked as $\eta_{adj}$ if they are adjacent to the center cell, highlighted by the dashed black line in the left and right panels. In this case, the cell is superelevated and enjoys a gradient advantage to only one other cell (immediately downdomain) and can thus avulse into this cell. The subscript "low" applies to all labeled cells but is omitted for legibility. Model representation of **(c)** abandoned channel repulsion and **(d)** attraction.

Abandoned channels on the floodplain can short-circuit this timescale by reducing the amount of aggradation needed to superelevate (Fig. 3A). If an abandoned channel is close to the active one, then this should encourage avulsion because during high flow there would be a steep water surface gradient that would cause erosion of the intervening levee and reroute the flow. This requires that the abandoned channel is roughly the same size as the active one and that it is close enough to increase the water surface gradient. What constitutes 'close enough' is unknown, though the Taquari avulsion is observed to proceed into an adjacent topographic low (~1 km from the parent channel; Fig. 2B), as are repeated avulsions along the Rapulo river (Edmonds et al., 2022). In effect, the lower elevation of the abandoned channel bed relative to its surrounding floodplain reduces the amount of aggradation needed for superelevation. We can thus rewrite the superelevation timescale of Eq. (1) as

$$T_A = \frac{\beta * \left( \eta_{adj,low} - \eta_{chan,low} \right)}{A_{chan} - A_{fp,tot}}, for\ \eta_{adj,low} > \eta_{chan,low} \tag{2}$$

where $\eta_{chan,low}$ and $\eta_{adj,low}$ represent the elevations (meters) of the active channel bed and the area adjacent to the channel, respectively (Fig. 3A,B). This adjacent elevation can vary based on the topography adjacent to the channel. For example, if there is an abandoned channel bed that is inset into the surrounding floodplain adjacent to the river, then the active channel becomes superelevated relative to the abandoned channel when $\eta_{chan,low} = \eta_{adj,low}$. When this difference $\eta_{adj,low} - \eta_{chan,low} < h_{chan}$ (see Fig. 3A), then Eq. (2) will result in a shorter avulsion set-up timescale than would be expected for a featureless floodplain (Eq. (1); Mohrig et al., 2000). Even though this is a simple amendment to Eq. (1), as we show later it has important effects on avulsion timing and location.

Channel reoccupation could also shorten superelevation timescales. Given the density of channels we observed on megafans (Fig. 1; Fig. 2), and observations from the stratigraphic and remote sensing records, it seems that reoccupation must be common (e.g. Mohrig et al., 2000; Chamberlin and Hajek 2015, 2019; Valenza et al., 2020). When active channels avulse, any previous aggradation downstream of the avulsion locus is not immediately destroyed. Instead, if these channels are later reoccupied, and had not been completely scoured out in the interim, then Eq. (2) allows for superelevation to be inherited. In these two ways, abandoned channels can cause rivers to have avulsion set-up timescales that are much less than via relative aggradation alone as embodied in Eq. (1).

## 4. Model conception and implementation

### 4.1 Model overview & routine

The prevalent channelization of fluvial megafan surfaces led us to consider how abandoned channels may affect avulsion dynamics and landscape evolution in foreland basin settings. To test these effects, we created a physically based cellular model of an evolving alluvial landscape with parameterized and tuneable abandoned channel dynamics ('RiverWalk'; available at DOI:10.5281/zenodo.5576789). Our model is intentionally simplified as much as possible while retaining the ability to recreate the essential features of fluvial megafans in foreland basins (Bokulich, 2013). As a brief conceptual overview, our model consists of a single river exiting a mountain-front and transporting some fixed amount of water discharge and sediment flux. As it enters a foreland basin, relative subsidence (high near the mountain front, decreasing linearly into the basin) causes sediment to be deposited preferentially near the mountain front. This leads to river avulsion via superelevation, and over time these avulsions construct a radially oriented fan through the emplacement of channels that individually aggrade before abandonment. In our model, these abandoned channels can affect avulsion dynamics. For simplicity, we ignore the impacts of other rivers or fans and of any other mountain-front processes that may advect sediment into the basin. The model is generally insensitive to small changes in most non-experimental parameters (Sect. 6.1).

The model routine operates as follows; more details on individual components are provided in Tables 1 and 2 and in the sections that follow. We paired a 1D diffusive channel-bed-elevation model (Paola et al., 1992) that describes how elevation in a river channel diffuses due to sediment transport with a rectangular, 2D cellular computational domain of 150 km per side

that describes the floodplain and surrounding basin. Following Jerolmack and Paola (2007), each cell has a low (channel-bed)
and high (levee or alluvial ridge) elevation. The simulation initializes by assuming the channel takes a straight path to the
bottom of the domain (Table 1). The 1D sediment transport model is calculated to equilibrium along this path (Table 1). This
profile is then used to initialize floodplain cells by setting the elevation of every floodplain cell equal to the equilibrium
elevation an equal distance from the mountain-front along this path. This creates an underfed basin because nearly all
subsequent river paths will be longer than a straight line, which causes aggradation and avulsion.
**Table 1: Cellular model parameters. Values for parameters were chosen to be representative of rivers commonly found atop**
**megafans in mountain-front regions, including those seen in Figure 1.**

| Parameter | Value |
|---|---|
| Timestep | 10 yr |
| Grid dimensions | 301 cells x 301 cells |
| Cell size | 500 m x 500 m |
| Random walk weights | In descending order of steepness: 40%, 27.5%, 17.5%, 10%, and 5% |
| Minimum superelevation ($\beta$ *sensu* Mohrig et al., 2000.) | Channel-base equal to neighboring floodplain cell ($\beta$ = 1) |
| Overbank aggradation (base rate; $A_{fp,base}$) | Upstream boundary: 2 x $10^{-7}$ m/yr<br>Downstream boundary: 5 x $10^{-6}$ m/yr |
| Subsidence rate (linearly interpolated; $\sigma$) | Upstream boundary: 1 x $10^{-5}$ m/yr<br>Downstream boundary: 5 x $10^{-6}$ m/yr |
| Initialization length | Variable; Set such that initial apex elevation is ~5-10% less than final apex elevation; see Table 3 |

After the first avulsion, a new river pathway is established within a single timestep from the avulsion point (Sect.

4.3.1) and is set one channel depth below the surface. The pathway is selected via steepness-weighted random walk to any
point along the bottom boundary of the domain (Sect. 4.3.1), and all floodplain cells along this path are converted to active
channel cells (Sect. 4.3). The timestep increments and the elevations of each cell along the new pathway are transiently diffused
to represent river adjustment (Sect. 4.3). At the upstream boundary of the diffusion model, water and sediment come in at a
fixed rate so that the surface slope does not change (Table 2), and at the downstream boundary the channel-bed elevation is
fixed at 0 m. Diffusion continues until an avulsion trigger (with a fixed probability at each timestep) occurs and avulsion
criteria (superelevation and gradient advantage) are satisfied for at least one active channel cell (Sect. 4.3.1). The avulsion
location is randomly selected from among viable cells and pathfinding proceeds as before, but now the river can be repelled
or attracted (i.e., captured) by abandoned channels. Pathfinding stops when the avulsion is successful and encounters the
bottom boundary, or when the avulsion fails after becoming terminally trapped (Sect. 4.3.1). In both situations the timestep is
incremented, but in the successful case any active channel cells that are no longer occupied become abandoned channel cells,
and in the failure case the domain is restored to its pre-avulsion state.
**Table 2: Sediment diffusion calculation parameters.**

| Parameter | Value |
|---|---|
| Initial specific discharge (apex) | 1.9 x $10^5$ m²/yr |
| Incoming sediment supply | 400 m³/yr |

| | |
|---|---|
| Basin width (for discharge calculation) | $5 \times 10^4$ m |
| Coefficient $A$ | 1.00 |
| Nondimensional coefficient of friction | 0.01 |
| $C_0$ | 0.7 |
| $S$ | 1.65 |
| $\rho_{sediment}$ | $2.65 \times 10^3$ kg m$^{-3}$ |
| $\rho_{water}$ | $1.00 \times 10^3$ kg m$^{-3}$ |


In all future timesteps, after updating the 2D landscape and before checking for avulsion triggers, floodplain and

abandoned channel processes routines are executed. First, cells experience subsidence at a rate that decreases away from the
mountain front (representing a foreland basin) and overbank floodplain deposition that varies with distance from the mountain
front but not with distance from the channel (Sect. 4.3.2). Next, abandoned channels are healed by a steady-rate topographic
adjustment function until they reach a specified healing endpoint (Sect. 4.3.2). Finally, any abandoned channel cells with less
than 25% of a mean channel depth in remnant relief are converted to floodplain cells (Sect. 4.3).

### 4.2 1D diffusive channel-bed elevation model

The 1D model has a variable length that is equal to that of the planform river pathway established in the 2D model.

We used transient diffusion to model channel-bed elevation changes along this pathway that would occur from sediment
transport (Paola et al., 1992):
$$\sigma + \frac{\partial \eta_{chan,low}}{\partial t} = \frac{\partial}{\partial x}\left(\nu \frac{\partial \eta_{chan,low}}{\partial x}\right), \qquad \nu = -\frac{8qA\sqrt{c_f}}{C_0(S-1)} \tag{3}$$

where $t$ is time (years), $x$ is space (meters), $\nu$ is diffusivity (square meters per year), $q$ is normalized water discharge per unit
basin width (square meters per year), $A$ is a non-dimensional constant set to 1, $c_f$ is a dimensionless drag coefficient, $C_0$ is bed
sediment concentration, and $S$ is sediment specific gravity ($\frac{\rho_{sediment} - \rho_{water}}{\rho_{water}}$, non-dimensional; Table 2). We used the Crank-
Nicolson solution scheme to solve this equation. This scheme is second-order, implicit in time, and unconditionally
numerically stable for diffusion partial differential equations (Slingerland and Kump, 2011). Treating diffusion of the bed
surface transiently (rather than bringing the river completely to equilibrium between each timestep [cf. Jerolmack and Paola,
2007]) allows for local aggradation or incision to occur on channel profiles out of equilibrium.

Our experimental design necessitated using nondimensional repulsion and attraction factors that are normalized to

channel depths. As such, it was necessary to determine channel depth ($h_{chan}$) for each active channel cell. We solved for this
at every active channel cell once per timestep following Paola et al. (1992; Table 2). This method allows depth to vary as a
function of local slope. Immediately after avulsion, slope variations along channels can be extreme. These extreme variations
in slope create unrealistic variations in depth over short distances. As such, when solving for channel depth, we bound
maximum and minimum slope to within a factor of two compared to the equilibrium profile.

### 4.3 2D cellular model: avulsions and floodplains

The computational domain is discretized into square cells of length 500 m. There are three types of cells in our model: active channel ($_{chan}$), abandoned channel ($_{aban}$), and floodplain ($_{fp}$). All cells have two elevations ('high' and 'low') that we track throughout each run. All elevations are measured in meters.

Active channel: Active channel cells represent the current pathway of the river. There is one contiguous pathway for flow per timestep. We selected a cell size such that modeled rivers are approximately one fifth of the width of a cell; as a result, channel-scale processes (like meandering, crevasse splays, or other lateral-distance-dependent depositional effects) are not resolved.

The low elevation in each active channel cell represents the channel bed and is updated by transient diffusion as described in Eq. (3). Then, high elevations are set to the greater of i) the high elevation at the last timestep, or ii) one channel depth above the bed, such that:

$$\left(\eta_{chan,low}\right)_t \text{ is given by Equation 3} \tag{4a}$$

$$\left(\eta_{chan,high}\right)_t = max\begin{cases}\left(\eta_{chan,high}\right)_{t-1} \\ \left(\eta_{chan,low}\right)_t + h_{chan}\end{cases} \tag{4b}$$

where $t$ is the current timestep and $t-1$ is the prior one. This assumes that an aggrading river constructs levees that can contain its flow depth, but levees are not lowered if the river incises.

Other cell types become active channel cells whenever they are occupied by the active channel after an avulsion. During this process, the low elevations of the new channel pathway are inset one channel depth down from the high elevations unless there is a channel that is already incised beyond this depth. This rule allows for channels to inherit levees (and superelevation) and does not further erode abandoned channel cells that are already incised more than one channel depth below their levees.

Abandoned channel: Abandoned channel cells include any cell that was once active but no longer contains water. These cells are still capable of attracting and repulsing pathfinding avulsions. Each cell has low and high elevations that reflect abandoned channel beds and levees, respectively. These elevations experience a linear healing rate that depends on healing mode but ultimately adjusts the channel bed and levee elevations toward a specified endpoint (Sect. 4.3.2):

$$\eta_{aban,low} = \left(\eta_{aban,low}\right)_{t-1} + \left(A_{fp,tot} - \sigma\right) + H_{low} \tag{5a}$$

$$\eta_{aban,high} = \left(\eta_{aban,high}\right)_{t-1} + \left(A_{fp,tot} - \sigma\right) + H_{high} \tag{5b}$$

where $A_{fp,tot}$ is the total overbank aggradation rate on the floodplain (meters per year; Sect. 4.3.2), and $H_{low}$ and $H_{high}$ are the healing rates (meters per year) applied to the low and high elevations, respectively.

Abandoned channel cells can become active channel cells if they are later occupied after an avulsion. Otherwise, they will become floodplain cells when:

$$h_{aban} < \left(0.25 * \bar{h}\right) \tag{6a}$$

$$h_{aban} = \left(\eta_{aban,high} - \eta_{aban,low}\right) \tag{6b}$$

where $\bar{h}$ is mean channel depth (meters) calculated over the entire length of the active channel at each timestep. While healing gradually lowers $h_{aban}$, there is no process that can increase this relief other than revisitation by the active channel, in which case the cells will become active channel cells.

Floodplain: Floodplain cells are those never been visited by a channel or have completely healed after visitation. High and low elevations are equal for floodplain cells except if they were once abandoned and have transitioned to floodplain (via the threshold in Eq. (11)) they maintain their unequal elevations until healing is complete. Floodplain cells do not repulse or attract pathfinding avulsions. However, their remnant (and possibly unequal) elevations do affect set-up and avulsion pathfinding via weighted random walk (Sect. 4.3.1).

Floodplain cells that retain any remnant relief are subjected to healing in the same manner as abandoned channel cells:

$$\eta_{fp,low} = \left(\eta_{fp,low}\right)_{t-1} + \left(A_{fp,tot} - \sigma\right) + H_{low} \tag{7a}$$

$$\eta_{fp,high} = \left(\eta_{fp,high}\right)_{t-1} + \left(A_{fp,tot} - \sigma\right) + H_{high} \tag{7b}$$

**4.3.1 Avulsion processes:**

Avulsion set-up: Avulsions occur via three steps: i) set-up, ii) initiation via triggering, and iii) floodplain pathfinding. Avulsion set-up (Slingerland and Smith, 2004) occurs from a combination of superelevation and flowpath gradient advantage. A cell is superelevated if the elevation of its channel-bed is equal to or greater than at least one of its five neighboring cells (not including the three upstream cells; Fig. 3A,B) by some fraction of a mean channel depth:

$$\left(\eta_{chan,low} - \eta_{adj,low}\right) \geq (\beta - 1) * \bar{h} \tag{8}$$

We set $\beta = 1$, which requires the active channel bed to meet or exceed an adjacent cell's low elevation (Mohrig et al., 2000). As such, cells are considered superelevated when:

$$\left(\eta_{chan,low} - \eta_{adj,low}\right) \geq 0 \tag{9}$$

Our results are insensitive to values of $\beta$ between 0.5 and 1. In addition to superelevation, an avulsion in our model must have a local gradient advantage over its previous pathway. We calculate this gradient over the first step into surrounding cells, as opposed to over the entire pathway (cf. Ratliff et al., 2018).

Avulsion triggering: Once a portion of a river is superelevated, some triggering event is necessary to initiate an avulsion. Predicting natural triggers is challenging because they can take the form of floods, ice damming, bank erosion, woody debris dams, neotectonics, meander bend cutoffs, beaver dams, bar migration, or other events that allow flow to escape normal channel confinement (Harwood and Brown, 1993; Smith et al., 1998; Ethridge et al., 1999; Jones and Schumm, 1999; Mohrig et al., 2000; Slingerland and Smith, 2004; Gibling et al., 2010, Morón et al., 2017). With that said, we know that trigger recurrence can only be as long as observed avulsion periods in natural river systems, which range from $10^1$ years on the Kosi River megafan to $10^3$ years on the Mississippi delta (Wells and Dorr, 1987; Aslan et al., 2005; Jerolmack and Mohrig, 2007).

We set an average avulsion trigger period of 30 years by specifying a fixed probability of a trigger occurring on any given
timestep. We select 30 years as it provides ample opportunity for a river to avulse, provided avulsion set-up criteria are met.
Since triggers cannot initiate avulsions in the absence of set-up via superelevation (Slingerland and Smith, 2004), this
effectively sets a lower limit on avulsion period, but the actual period may be longer if there are no superelevated river segments
along the active channel when a trigger occurs.

Avulsion pathfinding: Whenever an avulsion trigger occurs, avulsion pathfinding initiates from a randomly selected

active channel cell that meets the set-up criteria. From here, the new channel path follows a steepness-weighted random walk
if it remains in floodplain cells. Each step, the pathfinding avulsion can move into one of five cells (three downstream and two
lateral). The cell is selected randomly, and the choices are weighted by steepness (see Table 1 for weighting scheme). Model
outcomes are not sensitive over reasonable ranges of steepness weights, so long as all five directions are possible. The river is
prevented from returning to its previous position and movement beyond the domain boundaries.

When a pathfinding avulsion is adjacent to an abandoned channel cell, the model checks to see if the abandoned

channel cell is repulsive or attractive (Fig. 3C,D). Abandoned channel cells are repulsive when their levee heights above the
adjacent floodplain ($L_h$; meters) are larger than some multiple of the pathfinding avulsion flow depth ($h_{avul}$; meters):

$$L_h > \alpha_R * h_{avul} \tag{10a}$$

$$L_h = \left(\eta_{aban,high} - \eta_{appr,low}\right) \tag{10b}$$

where $\alpha_R$ is a nondimensional repulsion factor, $h_{avul}$ is the threshold channel depth calculated with diffusion theory (Paola et
al., 1992) assuming the flow is channelized during pathfinding, and $\eta_{appr,low}$ is the low elevation in the adjacent cell from
which the pathfinding avulsion channel approaches the abandoned channel. $\alpha_R$ is a threshold for how tall levees must be to
repulse advancing flow. Lower values are more repulsive since the threshold to repel is lower. A value of zero means that any
positive value of $L_h$ would cause repulsion.

Abandoned channel cells are attractive when $h_{aban}$ (meters) is larger than some fraction ($\alpha_A$) of mean flow depth:

$$h_{aban} > \alpha_A * \bar{h} \tag{11}$$

$\alpha_A$ is a threshold value describing how much remnant relief an abandoned channel must retain to capture flow. Lower values
are more attractive since it means only a small fraction of the original channel relief is required to be attractive. If captured,
the pathfinding avulsion will move in the direction of the lowest $\eta_{aban,low}$. This will continue unless there are no abandoned
channel cells into which flow can proceed, which can happen if the abandoned channel is discontinuous (Fig. 2), in which case
the river is ejected back onto the floodplain and resumes steepness-weighted random walk.

Rivers that are repulsed or not captured by abandoned channels will proceed via steepness-weighted random walk

until they exit the domain. If during pathfinding there are no viable moves, which can happen within floodplains bounded by
abandoned channels that cannot be reoccupied (Fig. 2), the avulsion fails, all cells are reverted to their pre-avulsion states, and
the model increments to the next timestep. While this implementation of failed avulsion pathfinding is a simplification, it
conceptually reflects healed crevasse splays (Slingerland and Smith, 2004) and matches limited observational evidence,
including among avulsions on the Rapulo river (Edmonds et al., 2022).

### 4.3.2 Floodplain processes:

Floodplain processes are applied to all abandoned channel and floodplain cells. These processes include rules for 1)
overbank deposition; 2) subsidence; and 3) abandoned channel healing.
Floodplain deposition & subsidence: We implement an overbank deposition rate that is constant along grid rows. As
channels are considered small relative to the width of a cell, we assume that any distance-from-channel-dependent component
to overbank sedimentation is contained within a single cell (cf. Bridge and Leeder, 1979; Pizzuto, 1987). Instead, and similar
to Jerolmack and Paola (2007), the total floodplain aggradation for each row ($A_{fp,tot}$; meters per year) is the product of a base
rate ($A_{fp,base}$, meters per year) and an additional term that increases linearly with the vertical distance between the highest
elevation ($\eta_{high,max}$) in that row and the elevation of a far-field floodplain cell that has never been visited by the active channel
($\eta_{farfield}$). While simple, this depth-dependent scaling reflects a basic intuition that regions of the basin that are inundated to
a greater depth beneath the highest levee (often the active channel) during flooding should receive more overbank sediment.
The vertical distance term is nondimensionalized by dividing by mean channel depth as averaged over the entire active channel.
The base rate $A_{fp,base}$ increases downstream, described by a linear interpolation between an upstream and downstream
boundary value, and reflects an increase in suspendable sediment (e.g., washload) downstream. Finally, we assume that total
overbank deposition on the floodplain ($A_{fp,tot}$) cannot exceed subsidence ($\sigma$, meters per year):

$$A_{fp,tot} = \min \begin{cases} A_{fp,base} * \dfrac{\eta_{high,max} - \eta_{farfield}}{\bar{h}} \\ \sigma \end{cases} \qquad (12)$$

Equation (12) deposits equal amounts of sediment on abandoned channel lows as highs, and thus does not heal
abandoned channels over time. Healing is handled separately and described in the following section. Finally, as a basic
approximation of foreland basin style subsidence, we apply subsidence at each timestep at constant rates. These rates vary
spatially via linear interpolation between a pair of rates representing proximal and distal values, with the proximal rates being
two times greater.
Abandoned channel healing: Despite the critical importance that floodplain topography and abandoned channel
healing timescales play in affecting channel network evolution in avulsing systems (Jerolmack and Paola, 2007; Reitz et al.,
2010), there is no consistent choice of rules for implementing this phenomenon in models of avulsion. In our model, we
implemented different abandoned channel healing styles to explore how they influence avulsion dynamics and landscape
evolution. Within these styles, abandoned channels can be healed 'bottom-up' as they are filled with sediment, 'top-down' as
their levees are eroded, or have both elevations adjusted toward the far-field floodplain (Fig. 4).
All healing modes adjust high, low, or both elevations linearly until a given endpoint is reached (Fig. 4). The healing
rates are set to

$$H_{high} = \alpha_{H,high} \frac{\bar{h}}{h_T} \tag{13a}$$


$$H_{low} = \alpha_{H,low} \frac{\bar{h}}{h_T} \tag{13b}$$


where $\alpha_{H,high}$ and $\alpha_{H,low}$ are the healing rate parameters and have values that range from -1 to 1, and $h_T$ is the characteristic
time needed to heal one mean channel depth, which we set as 55,000 years. The value of $h_T$ is necessarily speculative due to
the lack of observational data on healing rates. We came to our value by first estimating the fastest reasonable timescale over
which an $\boldsymbol{O:}10^0$ m deep abandoned channel (e.g., oxbow lake) can be filled when it is hydraulically connected to frequently
flooding rivers (e.g. Cooper and McHenry 1989; Wren et al., 2008). This yields a minimum $h_T$ of $\boldsymbol{O:}10^2$-$10^3$ yr. Abandoned
channels must almost certainly heal slower than this rate, as most abandoned channels are distant from the active channel at
any given time (Figure 1, 2), and net sediment deposition rates are known to decrease as observation window duration increases
(Sadler 1981; Schumer and Jerolmack 2009). Next, we estimated an upper limit to $h_T$ as the equilibration timescale of an
abandoned channel via diffusion alone, which is on the order of $\frac{L^2}{v_{fp}}$ (Paola et al., 1992). For our case, $L$ is a half-channel width
(~50 meters) and $v_{fp}$ can be approximated by hillslope diffusivity values (~0.005 square meters per year; Richardson et al.,
2019). This yields a maximum $h_T$ of $\boldsymbol{O:}10^5$ years. Finally, we chose a representative $h_T$ between these two limits. Future work
is needed to determine the validity of this assumed timescale, especially considering the importance of abandoned channels on
affecting avulsion set-up and pathfinding. Regardless, $h_T$ is held constant between experimental runs, which instead vary only
the healing direction mode. The first healing mode (deposition-only) raises abandoned channel lows toward levee-tops, such
that $\alpha_{H,high} = 0$ and $\alpha_{H,low} = 1$. The second healing mode (erosion-only) lowers levees toward channel-bases, such that
$\alpha_{H,high} = -1$ and $\alpha_{H,low} = 0$. The third healing mode (far-field directed) adjusts abandoned channel highs and lows toward
the far-field floodplain elevation at rates of $\alpha_{H,high} = -0.5$ and $\alpha_{H,low} = -0.5$. In all cases, once topographic highs and
lows have achieved their final healing endpoints (Fig. 4), $\alpha_{H,high}$ and $\alpha_{H,low}$ rates are set to 0.

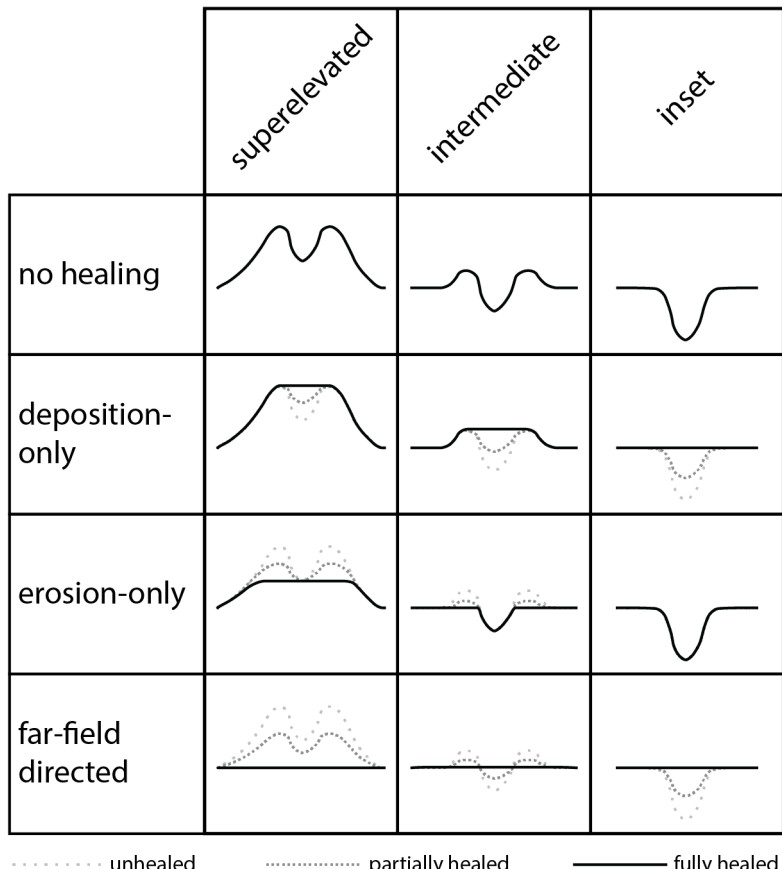


**Figure 4: Potential healing modes for different initial conditions of abandoned channels. Each healing mode has different endpoints depending on the initial channel emplacement: deposition-only adjusts each abandoned channel cell's low elevation toward its high elevation, erosion-only adjusts high elevations toward low elevations, and far-field directed adjusts both elevations towards the far-field floodplain elevation. As such, the deposition-only and erosion-only modes can result in topography that maintains positive topographic relief even once fully healed.**

**4.4. Experimental design**

We ran four series of model experiments to investigate how abandoned channel attraction, repulsion, and healing influence avulsion dynamics. A summary of non-experimental and experimental parameters is provided in Table 3.

**Table 3: Model parameters used to generate figures**

| Run duration (Myr) | Avulsion trigger period (years) | Healing timescale ($h_T$, years) | Initialization length (meters) | Repulsion factor ($\alpha_R$) | Attraction factor ($\alpha_A$) | Healing mode | Figure # |
|---|---|---|---|---|---|---|---|
| 5 | 10 | 10,000 | 57,000 | 4.00 | 0.25 | Far-field directed | 5 |
| 1 | 30 | 55,000 | 122,500 | 4.00 | 0.25 | Far-field directed | 6,8,9 |
| 1 | 30 | 55,000 | 122,500 | 4.00 | 0.25 | Far-field directed, deposition-only, and erosion-only | 7 |
| 5 | 30 | 55,000 | 122,500 | -0.50 to 8.00 | 0.25 | Far-field directed | 10 |
| 5 | 30 | 55,000 | 122,500 | 8.00 | 0.00 to 2.00 | Far-field directed | 11 |
| 10 | 30 | 55,000 | 122,500 | 4.00 | 0.25 | Far-field directed, deposition-only, and erosion-only | 12 |


The first series consists of a single base run with $\alpha_R = 4$, $\alpha_A = 0.25$, and far-field directed healing. Setting $\alpha_R = 4$
means that flow is repulsed when levees are four times the height of the approaching flow; this allows some channels to be
repulsive and others to not. Setting $\alpha_A = 0.25$ allows channels to capture flow so long as they are deeper than ¼ of a mean
channel depth, consistent with flume experiments (Reitz et al., 2010) that show old, in-filling abandoned channels acting as
attractors with little remnant relief. For abandoned channel healing, we employed far-field directed healing because its endpoint
of a totally flat plane is equivalent to that of diffusion on a laterally infinite plane, approximating the effects of floodplain
diffusion without the computational cost.

Our second set of runs explored the importance of abandoned channel repulsion on where, when, and why avulsions
occur by varying $\alpha_R$ from -0.50 (most repulsive) to 8 (least repulsive), while holding $\alpha_A = 0.25$. Each run is a 5 Myr simulation
using the far-field directed healing mode. Next, a matching third set of runs was performed to investigate the effect of $\alpha_A$ by
varying it from 2.00 (least attractive) to 0 (most attractive) and setting $\alpha_R = 8$. Our final set of runs investigated the role of
abandoned channel healing mode without changing $h_T$ (Fig. 4). We hold $\alpha_A$ and $\alpha_R$ constant between each 10 Myr run.
**4.5 Analysis**

We analyzed the planform appearance of generated topography and the location of avulsions for each run. For figures
showing planform appearance (Fig. 5; Fig. 6; Fig. 10-12), we show each cell's high elevation normalized relative to the
$\eta_{farfield}$ for its row. We did this because megafans are low-relief features, and the change in elevation along dip otherwise
overwhelms the signal (Fig. 5). We quantified avulsion locations by recording the straight-line distance from the mountain-
front to each avulsion. These data were binned every 6.25 km and plotted as histograms showing the number of avulsions
moving away from the upstream boundary. These values are normalized to the bin with the greatest occurrence. For Fig. 6, we
measured and binned avulsion locations in the same way for a second run without relative superelevation, but normalized this
histogram to that of the base run to display the overall reduced number of avulsions. We also analysed avulsion locations by
creating smoothed (50 kyr moving window average) curves of recorded distance to the mountain-front that show how median
and 95th percentile (i.e., distal) avulsion locations change over the course of simulations. Finally, we analyzed differences
between the proximal and distal domains for our base run by tracking the along-strike position of the active channel at two
distances (12.5, 50 km) from the mountain-front for every timestep (Fig. 7).

## 5. Model results

### 5.1. Base run and validation

We validated our model results by comparing model output for our base run with megafan topography from ICESat-
2 (Neuenschwander et al., 2020) via the OpenAltimetry platform (Khalsa et al., 2020). ICESat-2 is a continuously measuring
(10 kHz, ~0.7 m between points on the ground) satellite that collects vegetation-penetrating laser altimetry (Neuenschwander
and Pitts, 2020). ICESat-2 offers greater precision than radar-derived elevation at the cost of limiting data collection to ~north-
south oriented linear tracks. While our model does not aim to precisely simulate any specific fan, the simplified model recreates
the essential features of mountain-front fluvial megafans. The 1D elevation diffusion model reproduces rivers with appropriate
channel depths and slopes, while the 2D cellular model recreates the broad, low-relief, convex-up fan shape typical of megafans
(Fig. 5). Along-dip comparisons on the Pucheveyem fan (Fig. 5) are also favorable, showing similar low-relief slopes (~$O$:10$^{-}$
$^{3}$). These slopes change abruptly at a topographic break marking the end of the fan topography, with a shallower gradient in
the distal domain.
The model produces two distinct domains despite no external parameters varying with distance to the mountain-front,
aside from a linearly decreasing subsidence rate and increasing overbank aggradation rate. Further, these two domains still
emerge within the model even when these two parameters are held uniform. The two domains generated are consistent with
earlier remote sensing observations (Fig. 1). The proximal domain is a zone of sediment distribution created by repeated
avulsions; it has a steeper slope (Fig. 5) and the abandoned channels that create the topography are radially distributive (Fig.
6A). In this domain, frequent channel avulsion causes small lateral adjustments to river position, filling local topographic lows
(Fig. 7A). Avulsion probability is highest at the apex because that is where sediment is introduced (Fig. 6B). In contrast, the
distal domain is a zone with a dominantly tributive geometry; it has a shallower slope (Fig. 5) and is much more sparsely
channelized (Fig. 6A). In this domain, the active channel switches between fewer, more-persistent channels (Fig. 6A; Fig. 7A).
Flow becomes confined to these more-persistent channels because avulsions that occur upstream are quickly captured and
routed into one of a finite number of pre-existing pathways (Fig. 7A). Distal abandoned channels that are occupied infrequently
can partially or fully heal between revisitations, creating discontinuous alluvial ridges (Fig. 2; Fig. 6). Avulsion probability
rapidly decreases past the fan boundary (Fig. 6B), and along-strike topographic relief is nearly flat, which compares favorably
to previous observations of megafans (Hartley et al., 2010a; Bernal et al., 2011). Notably, we reproduce these channel and
megafan features despite the absence of bounding rivers or other external topographic controls that are seen on some modern
megafans (Fig. 1E,F).

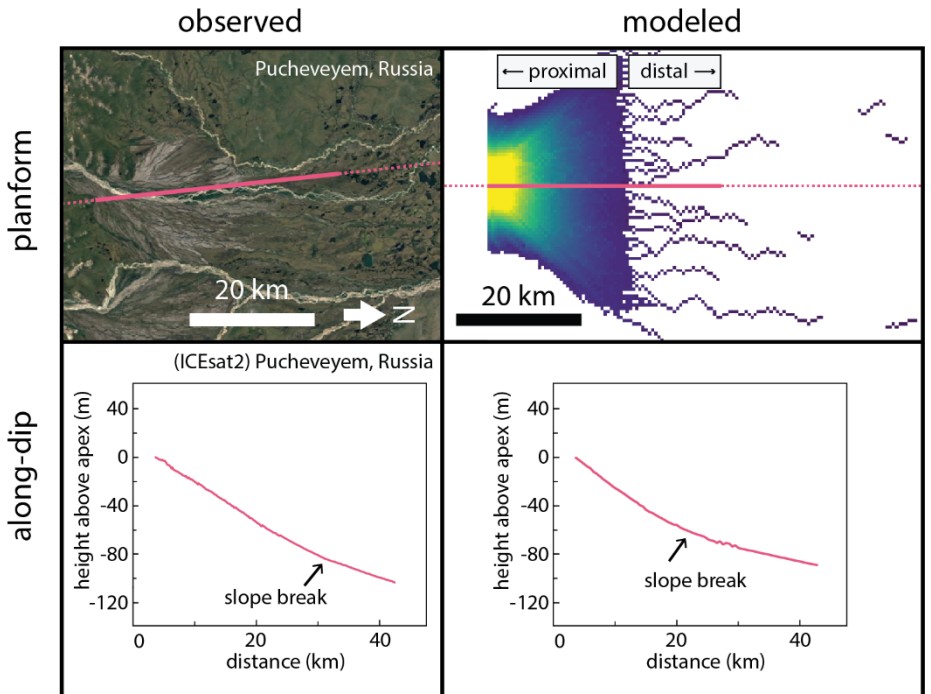


Figure 5: Megafan topography from model output compares favorably with real megafans, including delineation into distinct domains of slope. Colorbar and explanation for modelled planform is provided in Figure 6. Satellite images are USGS/NASA Landsat/Copernicus, © Google Earth.

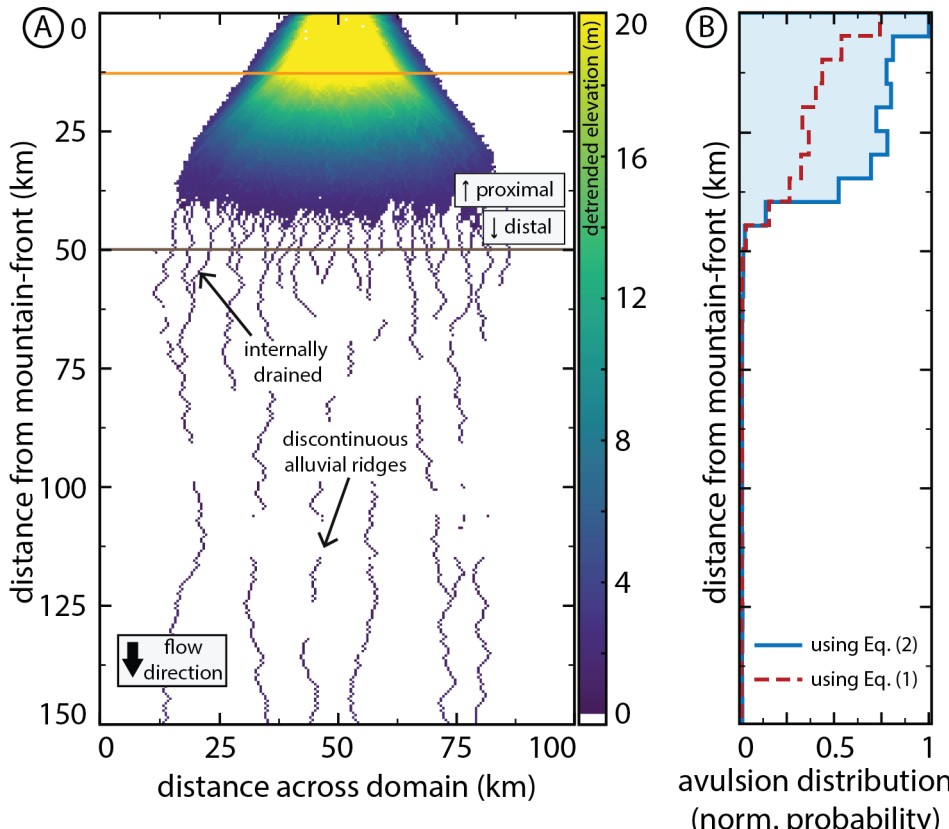


Figure 6: (a) Planform output of detrended high elevations from the base run. The colorbar is chosen such that negative or near-zero detrended values appear white. The location of the active channel is not shown. The model produces two distinct domains (proximal and distal) in addition to several marked features which compare well with observed megafans in the real world (Fig. 1). Orange and dark brown horizontal lines show the proximal and distal measurement locations, respectively, for Fig. 7A. (b) A histogram (bin-width 6.25 km) showing the downstream distribution of avulsion loci. Blue line corresponds to the model run shown in (A), whereas the dashed red line is an equivalent run that differs by requiring one full channel depth of aggradation to achieve superelevation (Eq. (1)) instead of measuring elevation relative to adjacent cells (Eq. (2)). The run using Eq. (2) had a mean time between avulsions of 32 years, compared to 57 years for the run using Eq. (1). Vertical axis scale for (b) is the same as (a).

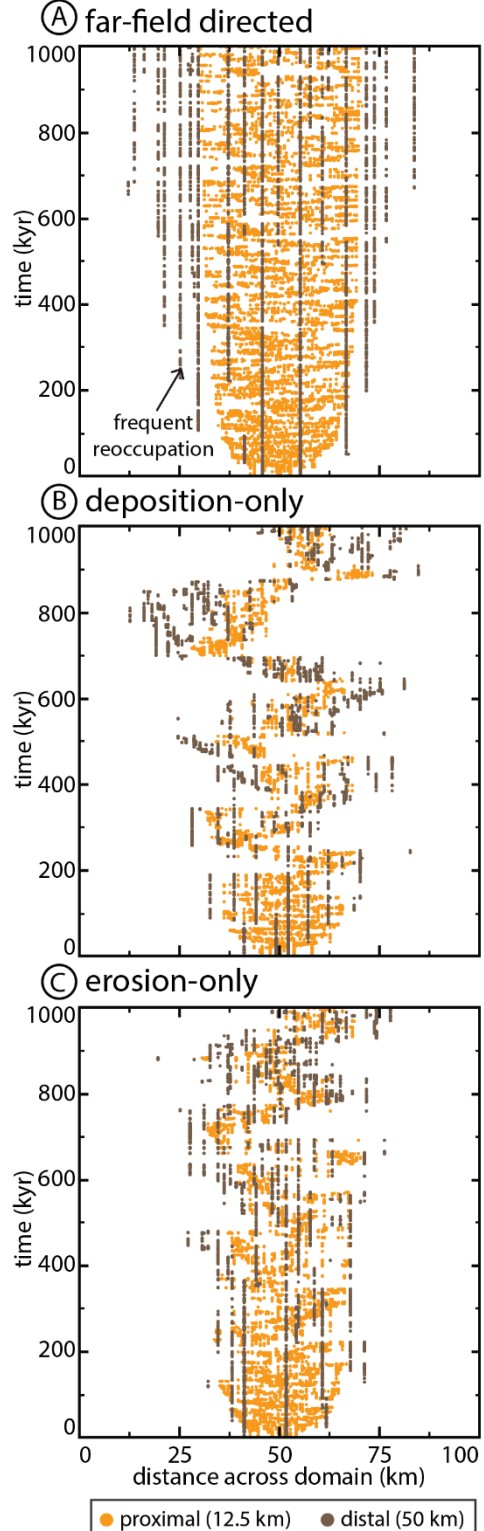

**Figure 7: Active channel position histories over 1 Myr at two distances from the mountain-front for three runs. Distances to the**
**mountain-front are illustrated via horizontal bars in Fig. 6A and Fig. 12A. In each run, only one channel position is possible per**
**timestep. (a) A run using the same parameters as Fig. 6. Note frequent and continued reoccupation for distal river positions. (b) and**
**(c) show runs identical to (a) except the healing modes are deposition-only and erosion-only, respectively (Fig. 12A). These runs show**
**similar behavior to (a) in early years but transition to lobe-switching behavior.**

## 5.2 How abandoned channels affect avulsion dynamics

Abandoned channels affect the timing and location of avulsions in four different ways: 1) superelevation shortcutting,
2) inheritance, 3) post-avulsion diffusion of the channel-bed, and 4) confluence aggradation. Each is discussed below.
We implemented avulsion set-up by measuring superelevation of an active channel relative to surrounding floodplain
topography (Eq. (2); Fig. 3). To investigate the effect of abandoned channels on this set-up, we performed an additional run
that is equivalent to our base run in Fig. 6 except for requiring each cell to aggrade a specified fraction ($\beta = 1$) of a channel
depth between each avulsion (Eq. (1); Fig. 6B). Compared to this run, the base run had a greater number of avulsions, especially
on the megafan surface downstream of the apex (Fig. 6B). Further, the run using Eq. (2) had a mean time between avulsions
of 32 years, compared to 57 years for the run using Eq. (1). Measuring superelevation relative to floodplains allows local
topographic lows associated with former abandoned channels to provide attractive locations for avulsion initiation, shortcutting
superelevation timescales (Jerolmack and Mohrig, 2007). Therefore, a densely channelized proximal domain generates
additional superelevation opportunities, spatially concentrating avulsions (Fig. 6B).
Abandoned channels also affect avulsion set-up indirectly through reoccupation mechanics. Superelevation is
inherited when avulsive flows reoccupy former abandoned channels. In a superelevated channel reach, an avulsion will strand
superelevated portions of the river that are downstream of the avulsion locus (discontinuous alluvial ridges in Fig. 6A). In this
way, avulsions can leave behind abandoned channels that may require minimal aggradation to achieve superelevation if they
are reoccupied before being healed, particularly if those channels are themselves adjacent to abandoned channel topography
that provides relative superelevation.

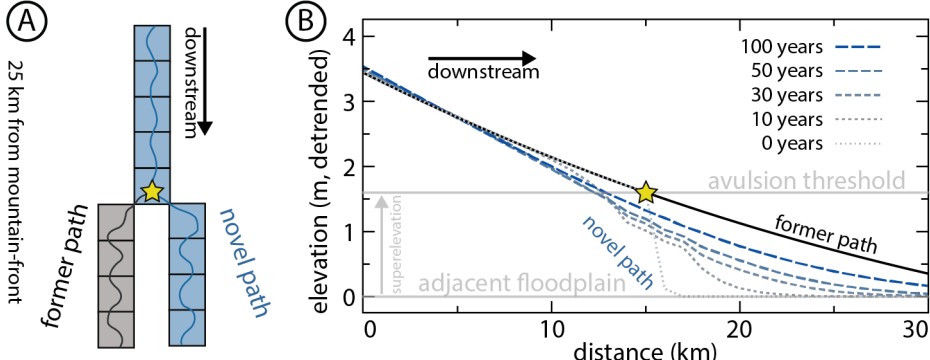


**Figure 8: Model example showing channel evolution immediately after an avulsion at a node marked by a star. (a) Planform arrangement of the parent channel and avulsion node, with both pathways of equal length. (b) Detrended elevation of the parent channel (relative to the adjacent floodplain) and new avulsion pathway upstream and downstream of the avulsion node. Immediately prior to the avulsion (black line), all cells upstream of the avulsion node are superelevated and are equally likely to avulse if a trigger occurs. After the avulsion, gradual knickpoint propagation upstream reduces superelevation. Downstream of the avulsion site, there is significant deposition that reduces the time to superelevation.**

In our model, avulsion set-up is also affected by local effects immediately after avulsions due to transient diffusion. This occurs in two ways. Firstly, superelevated cells upstream of the avulsion locus are not instantly lowered but instead require time for the knickpoint to propagate upstream (Fig. 8). In our simulation, the post-avulsion upstream reduction in channel bed superelevation proceeded gradually, migrating only several kilometers 100 years after an avulsion (Fig. 8). In this way, an avulsion does not instantly undo the avulsion set-up of cells upstream and future triggers can still cause avulsions to occur over this domain. Secondly, immediately downstream of an avulsion locus there is significant aggradation; a channel can diffuse nearly a meter of sediment into a downstream active channel cell within a decade (Fig. 8). In the case that these downstream cells are themselves already nearly superelevated, this can provide sufficient aggradation above the adjacent floodplain to set-up these cells. This effect is even more pronounced when new active channel cells are adjacent to abandoned channel lows, and thus have lower superelevation thresholds.

In our model, we observed abandoned channel confluences wherever a pathfinding flow is captured by a previous abandoned channel. Captured channels follow steepest-descent pathfinding within the network of occupiable abandoned channel cells. Within the distal, tributive domain, the number of possible abandoned channel pathways that can be occupied decreases with increasing distance from the mountain-front (Fig. 6A; Fig. 7A). This allows locations downstream of confluences to be more continuously occupied while the flow switches pathways upstream. This has important effects on avulsion because more aggradation occurs downstream of the tributary junction. Consider a scenario where avulsions on the fan always route flow into one of two possible paths (Fig. 9). The pathway downstream of the confluence is occupied 100% of the time while each parent pathway is occupied approximately half of the time. As channel-bed aggradation occurs only during active channel occupation, aggradation downstream of the confluence can therefore be greater than that observed in

either upstream pathway (Fig. 9C). As such, in the distal domain, abandoned channel reoccupation should preferentially focus

avulsions downstream of abandoned channel confluences.

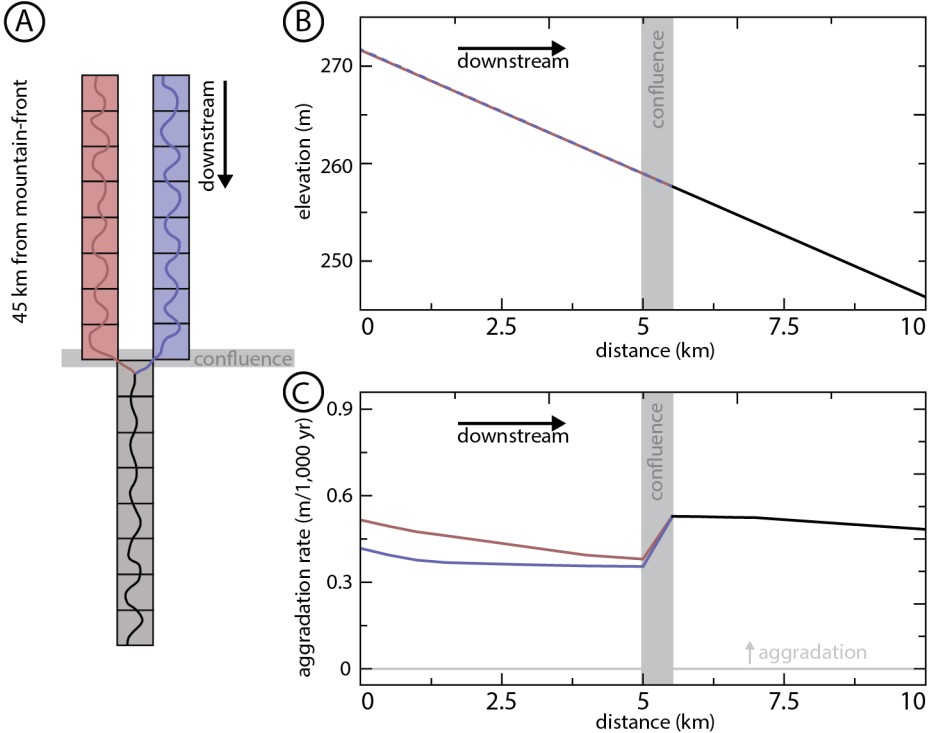

**Figure 9: Model experiment showing channel evolution at an abandoned channel confluence. (a) Planform arrangement where the channel avulses between the red and blue pathways whenever normal avulsion criteria are satisfied. (b) Elevations of the red, blue, and gray channel segments upstream and downstream of the confluence. These elevations are not detrended. The blue channel profile is dashed to not obscure the red channel profile. (c) Aggradation rates over a 3,000 year period along the three channel segments. Repeated avulsions mean that the red and blue channels alternate deposition, while the gray channel downstream is constantly occupied leading to a faster aggradation downstream of the confluence.**

## 5.3 Abandoned channel repulsion

We observed the effects of varying $\alpha_R$ on both planform appearance and the location of avulsions with a constant $\alpha_A$

(Fig. 10). Increasing repulsion (decreasing $\alpha_R$) extends the proximal domain farther downstream; increasingly repulsive runs
are increasingly distributive and generate fewer tributary confluences (Fig. 10A). Further, runs that are highly repulsive do not
generate the abrupt downstream change in avulsion frequency seen when $\alpha_R \geq 1.00$. Instead, highly repulsive runs show relative
avulsion frequencies that follow a power-law-like distribution with distance from the mountain-front (Fig. 10A).

The proximal domain propagates farther downstream when $\alpha_R$ is smaller (more repulsive) because avulsion location

propagates farther downstream (Fig. 10B). While all avulsion location curves show a downstream progradation of avulsion
locations during runs as the fan grows, both the median and 95[th] percentile shift downstream between runs with decreasing $\alpha_R$
when $\alpha_R \leq 1.00$ (i.e., where avulsive flows must be equal to or greater than levee heights above surrounding floodplains to
reoccupy). Median avulsion locations are less affected than 95[th] percentile curves, indicating that distribution skewness
increases.
Increasing repulsiveness (decreasing $\alpha_R$) pushes avulsions farther from the mountain-front because flow in the
proximal domain is concentrated into fewer channels, allowing for sediment (and therefore superelevation) to propagate farther
downstream (Fig. 10). As a contributing effect, runs with lower $\alpha_R$ create more internally drained basins that themselves cause
avulsions to fail. Since failed avulsions cause the timestep to increment without changing river positions, the time between
successful avulsions is greater in runs with many failed avulsions, and sediment can thus propagate farther along active
channels. This encourages channels in the distal part of the model to superelevate and avulse more often.

## 5.4 Abandoned channel attraction

Abandoned channel attraction dynamics also impact both avulsion locations and planform appearance during model
runs (Fig. 11). With a constant $\alpha_R$ and increasing abandoned channel attraction (decreasing $\alpha_A$), the transition from distributive
to tributive domains shifts up-domain and fan width increases (Fig. 11A). When $\alpha_A$ is large (low attractiveness), model output
resembles a series of weighted random walks because abandoned channels rarely capture flow and steepness weighted random
walk determines channel position. Like the repulsion simulations, both the median and 95[th] percentile avulsion locations are
affected by changing attraction parameters (Fig. 11B). Decreasing $\alpha_A$ (increasing attractiveness) pulls avulsions towards the
mountain-front, and the greatest change is for $\alpha_A$ between ~0.50 and 1.50. Minimal change occurs for $\alpha_A$ values above and
below this range. In contrast, when $\alpha_A$ increases (attractiveness decreases), the fan lengthens and avulsions occur farther down-
domain because fan surfaces host abundant abandoned channels that influence avulsion dynamics. This interpretation is
supported by the avulsion histograms, where low-attractiveness runs show a non-zero avulsion frequency plateau in the distal
reaches and a more gradual downstream reduction in frequency than in more-attractive runs (Fig. 11A).

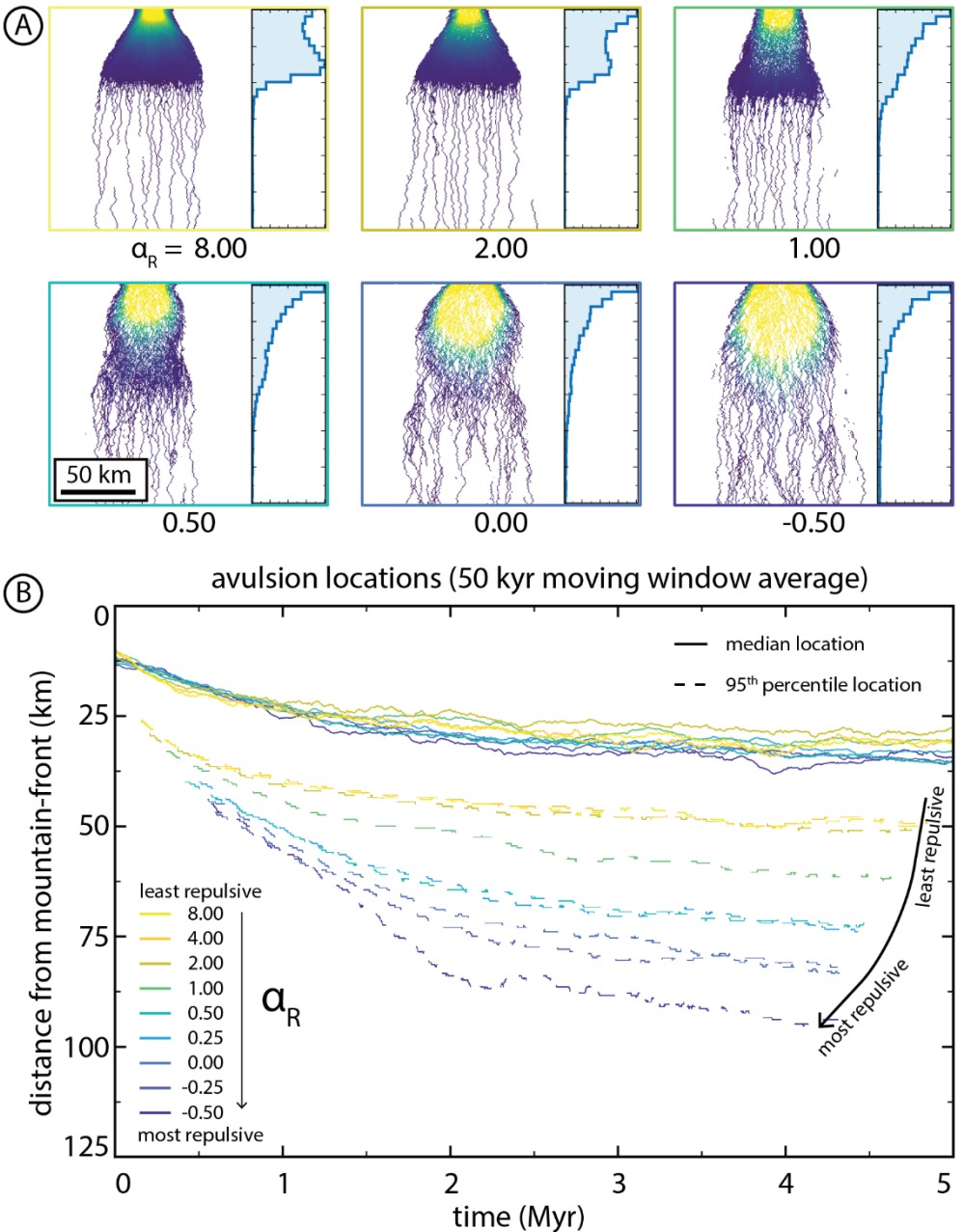


**Figure 10: The effect of abandoned channel repulsion on (a) planform appearances and normalized avulsion location histograms, and (b) median avulsion locations through time. Decreasing $\alpha_R$ causes avulsion location to move downstream. These changes are more pronounced for 95th percentile locations. Color scale for inset planform appearances is the same as in Fig. 6.**

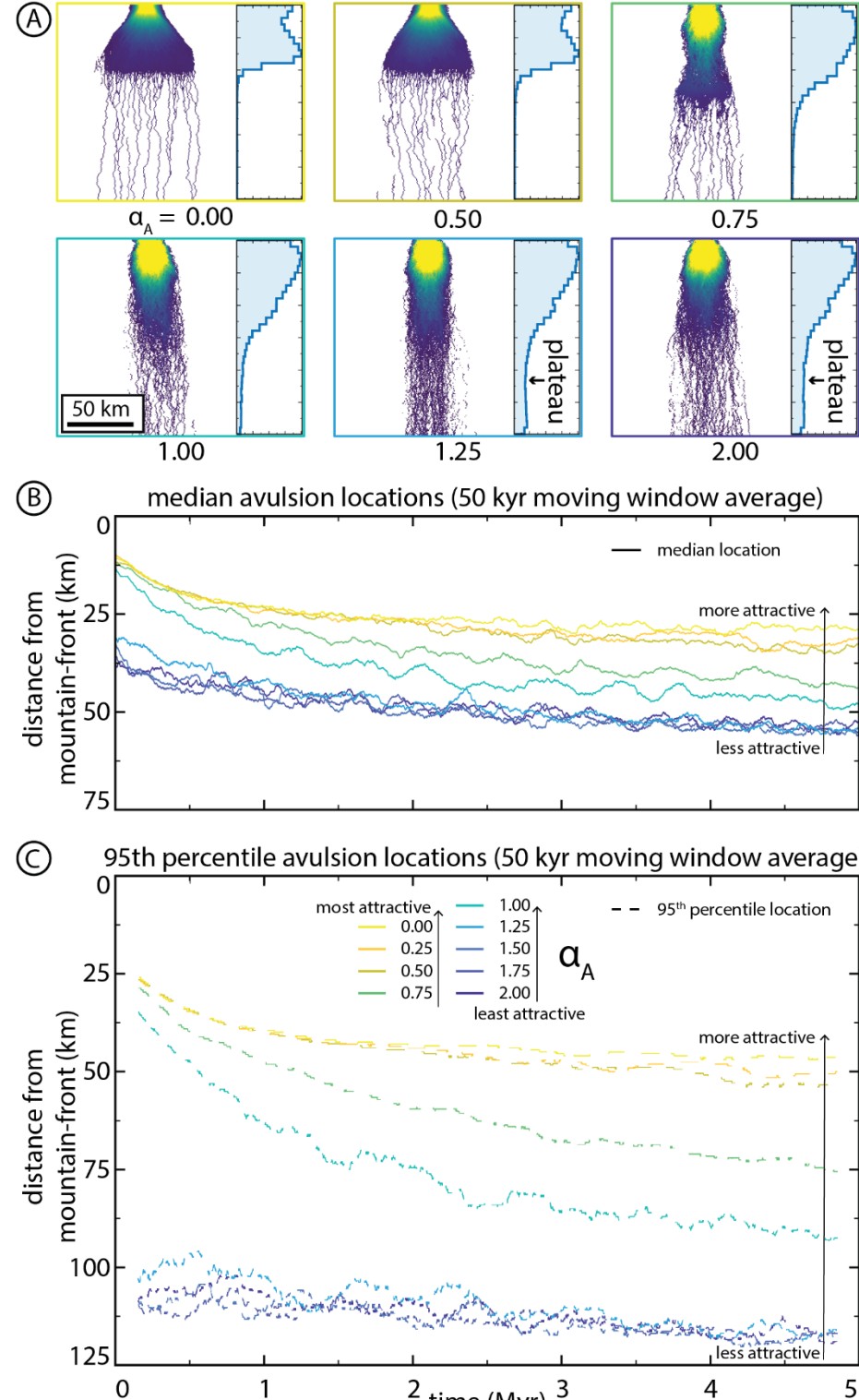

(A) $\alpha_A = 0.00$    0.50    0.75

1.00    1.25    2.00

50 km

plateau    plateau

(B) median avulsion locations (50 kyr moving window average)

distance from mountain-front (km)

— median location

more attractive ↑

less attractive

(C) 95th percentile avulsion locations (50 kyr moving window average)

distance from mountain-front (km)

most attractive ↑
0.00
0.25
0.50
0.75

1.00
1.25
1.50
1.75
2.00

$\alpha_A$

least attractive

− − 95th percentile location

more attractive ↑

less attractive

time (Myr)

**Figure 11:** The effect of abandoned channel attraction on (a) planform appearances and normalized avulsion location histograms, and (b,c) characteristic avulsion locations through time. $\alpha_A$ legend and x-axis scale in (c) applies to (b) as well. Note the difference in y-axis range between (b) and (c). Between $\alpha_A$ values of 0.50 and 1.50, increasing $\alpha_A$ causes predictable increases in the distance between the mountain-front and median and 95[th] percentile avulsion locations. These changes are more pronounced for 95[th] percentile locations, indicating greater skewness. Color scale for inset planform appearances is the same as in Fig. 6.

**5.5 Healing mode**

The healing mode determines how abandoned channels are gradually removed from the floodplain. By conducting 10 Myr base runs with different healing modes, we found that the deposition-only and erosion-only runs generated fans that nearly entirely filled up the simulation space over the course of several million years (Fig. 12). This occurs because the remnant topography of abandoned channels was never entirely removed by healing between visitations (Fig. 4). This is true even for the erosion-only run as, by definition, channels that achieve superelevation before abandonment have bases that are higher than surrounding floodplains. Since healing in erosion-only runs terminates once levees reach channel-beds, superelevated abandoned channel-beds on the floodplain remain indefinitely.

Healing mode affects avulsion location and introduces a new dynamic for fan growth. In erosion-only runs, the avulsion location propagates the farthest into the basin (Fig. 12B). Interestingly, the median and 95[th] percentile time series for deposition-only and erosion-only avulsion locations show spikes that represent avulsion location rapidly moving toward the mountain-front. These spikes represent lobe switching events, where avulsion loci shifted proximally as depositional space lower on the fan is filled and apical avulsions reroute flow to new regions on the fan surface (Fig. 7B,C; Supplemental Videos 1-3; DOI: 10.5446/54887). This compares well to observations of real-world megafans where deposition is interpreted to have occurred on discrete lobes (Chakraborty et al., 2010; Zani et al., 2012; Assine et al., 2014; Weissmann et al., 2015; Pupim et al., 2017). Lobe switching emerges in the model when deposition is localized in a particular region sufficiently long for a lobate area to become raised relative to other areas on the floodplain. This lowers the effective slope of this pathway, leading to a slope disadvantage over other regions on the floodplain. Future apical avulsions can then redirect flow to these other lower regions due to slope-weighted pathfinding, leading these lower regions to themselves eventually become raised and begin the cycle anew. Lobe switching does not occur during the earliest stages of fan growth because slopes are relatively steep on all faces of the fan and there is thus little intervening topography that could prevent an avulsing river from accessing other areas on the fan surface.

In contrast to the deposition-only and erosion-only runs, the far-field directed simulation achieved dynamic equilibrium relatively quickly and maintained a well-defined boundary between the proximal and distal domains for the remainder of the run. This occurs because it is the only healing mode that completely removes abandoned channel topography from floodplains. As such, this is the only healing mode that erases the topographic, attractive, and repulsive "memories" (*sensu* Reitz et al., 2010) of abandoned channels. Lobe switching on the same timescale is not observed in these runs because, unlike the deposition-only and erosion-only runs, far-field directed runs do not preserve topography indefinitely and alluvium is removed too quickly to build up regional slopes that can effectively resist pathfinding. Thus, while lobe switching seems to

have somewhat different frequencies based on whether abandoned channel healing is dominated by infilling or levee erosion,
the overall existence of lobe switching on this $O$:$10^5$ timescale is sensitive to the preservation potential of superelevated channel
beds and alluvial ridges.

It is important to note that both the erosion-only and deposition-only runs exhibited the typical separation of planform

space into two domains as they prograded, until the proximal domain encountered the edge of simulation space and the only
further adjustment that could occur was via vertical aggradation. Despite this, the deposition-only and erosion-only runs appear
to have less abrupt downstream avulsion frequency changes compared to the far-field directed one because the histograms are
time-integrated and reflect avulsion locations throughout the entire history of the run, including during progradation (Fig.
12A).

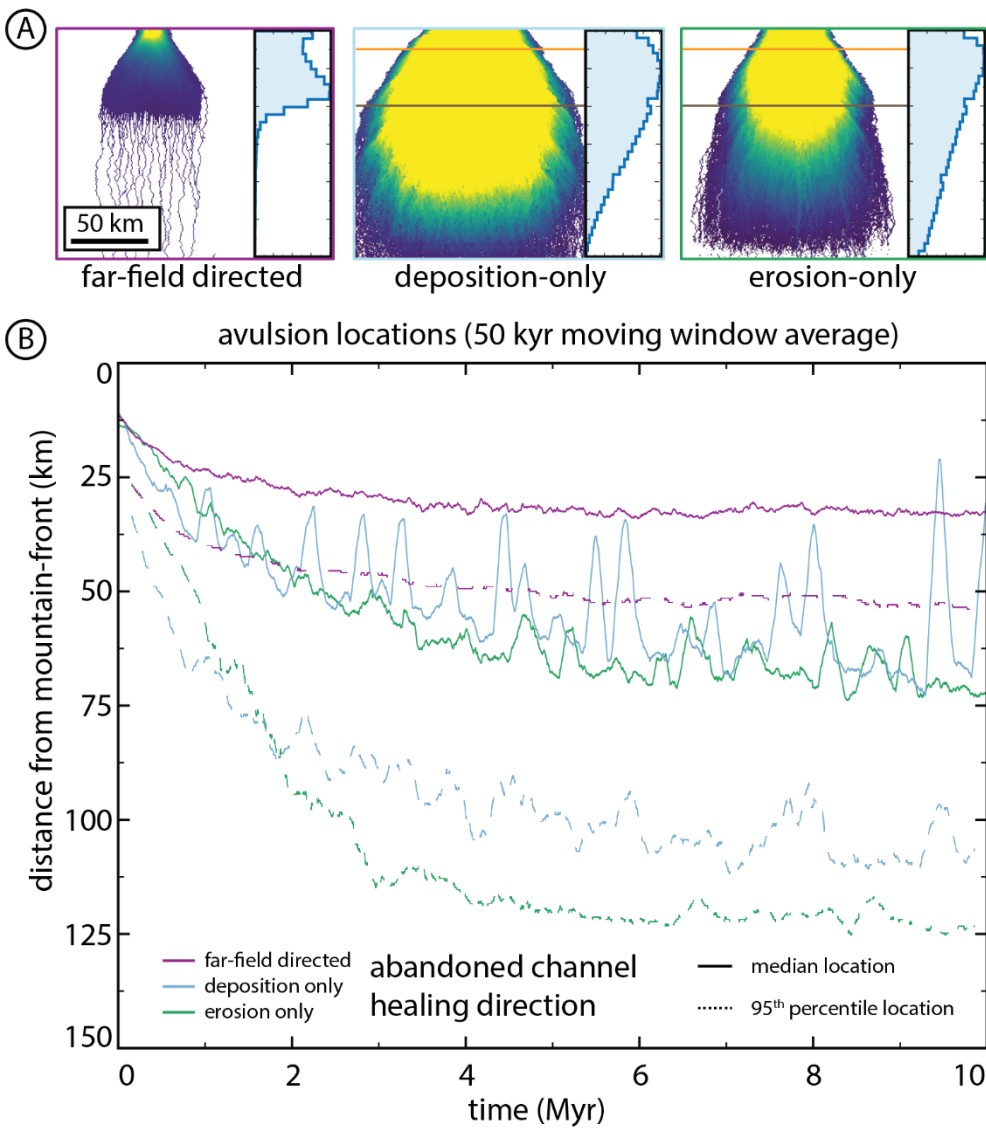


**Figure 12: The effect of healing mode on (a) planform appearances and normalized avulsion location histograms, and (b) characteristic avulsion locations through time. Runs are identical other than employing different abandoned channel healing modes. Orange and dark brown horizontal lines show proximal and distal (respectively) distances from mountain-fronts for Fig. 7B (deposition-only) and 7C (erosion-only). Color scale for inset planform appearances is the same as in Fig. 6.**

## 6. Discussion

### 6.1 Model sensitivity to non-experimental parameters

Our model is generally insensitive to small variations within reasonable ranges for most parameters presented in Tables 1 and 2, including values for random walk weights, minimum superelevations ($\beta$), overbank aggradation base rates $A_{fp,base}$, subsidence rates ($\sigma$), initialization lengths, abandoned channel healing timescales ($h_T$), and incoming specific discharge and sediment supplies. One exception is that the overbank aggradation rate and subsidence rate at the bottom boundary of the domain must be equal to satisfy the downstream boundary condition, and if subsidence is much larger than aggradation upstream, the basin can sag due to underfilling over stratigraphic timescales. We performed test runs with more functional changes, including runs that employed uniform subsidence (as opposed to foreland basin style subsidence), overbank deposition base values ($A_{fp,base}$) that did not change with distance to the mountain-front, or channel depths ($h$) and avulsion flow depths ($h_{avul}$) that did not vary along channel. In all cases, we still recreated the fundamental findings of two distinct emergent domains and the effects of abandoned channel repulsion, attraction, and healing on avulsion location.

### 6.2 Abandoned channels in avulsion models

Our model was designed to investigate the role of abandoned channels as both topographic repulsors and attractors during avulsion pathfinding. In doing so, we demonstrated effects on avulsion location and planform landscape evolution. We showed that abandoned channels affect the frequency and position of avulsions. Importantly, we demonstrated that the typical low-relief megafan with a transition from proximal (distributive, densely channelized) to distal (tributive, sparsely channelized) domains originates only when avulsion repulsion is infrequent and attraction is frequent (Fig. 10; Fig. 11). As such, when creating avulsion models, it is worth explicitly addressing abandoned channel creation, rate of healing, mode of healing, and interactions with future avulsion set-up and pathfinding because these factors fundamentally change avulsion dynamics and planform appearance of fluvial systems.

Previous stratigraphic models that simulate accumulation of channel bodies in a 2D strike-oriented cross-section have had to employ rules in order to emplace successive channels without resolving planform pathfinding (Sect. 2). These rules typically are random, compensation (lowest elevation), or clustered (channel emplacement occurs near previous channel location). These rules create important differences in simulated alluvial stratigraphy (Chamberlin and Hajek, 2015), however the floodplain conditions that lead to each rule are unknown. Our results show that the position of successive channels after avulsion follows different emplacement rules in proximal and distal domains for moderate attraction and repulsion (Fig. 7). In the proximal domain, avulsion pathways follow steepest descent that should generate compensational stratigraphy by seeking

local, not global, topographic lows (Fig. 7). This compares well to limited observational data showing that most avulsions
initiate into topographic lows and travel relatively small lateral distances before joining abandoned channels (Edmonds et al.,
2016; Valenza et al., 2020). In our deposition-only and erosion-only runs, emergent lobe switching provides an additional
process that can control channel positions, creating both clustering (channel switching within lobes) and compensation (lobes
switch to compensate; Fig. 7; Fig. 12). In the distal domain, channels are nearly perfectly clustered because flow routing
switches between a small number of active channels in a network, each of which can heal if they are not revisited for a sufficient
amount of time (Fig. 7). This compares better to the experimental model and flume observations of Jerolmack and Paola (2007)
and Reitz et al. (2010). As a caveat, when abandoned channels influence pathfinding, our model shows that it is not always
possible for avulsions to find the globally lowest point in the whole domain for a given cross-section (cf. Bridge and Leeder,
1979) because there may be high topography in-between that prevents pathfinding. This is particularly evident in the lobe
switching shown in Fig. 7B and Fig. 7C, where the global lowest point may exist outside of lobe deposition, but no viable
route exists to reach that point.
**6.3 Floodplain topography and evolution**
Our model shows that lobe switching on megafans only appears under certain abandoned channel healing rules (Fig.
7; Fig. 12). Floodplain topography, including abandoned channels, is thus a critical control on avulsion dynamics and landscape
evolution and modelers who wish to recreate foreland basin topography must be conscious of how they choose to implement
abandoned channel healing. While our results indicate that the preservation of abandoned channel topography between
avulsions is necessary for lobe switching to emerge, further research can be directed towards uncovering other necessary
conditions, and thus whether it is appropriate to assume that the presence of lobe switching on real world fans is a predictor of
abandoned channel healing mode. Regardless, the dependence of lobe switching on abandoned channel healing mode within
our model emphasizes Jerolmack and Paola (2007)'s identification of the remarkable lack of knowledge regarding the
competing processes of topographic construction and destruction on floodplains. The principal topographic features of
floodplains in aggradational (*sensu* Weissmann et al., 2015) settings appears to be abandoned channels, including both
topographic highs and lows (Fig. 2). Understanding the extent to which abandoned channels and floodplain topography control
avulsion dynamics in natural systems requires a better understanding of floodplain topography.
Given the extent of these unknowns, considerable insight about floodplain evolution could be gained from highly
detailed investigations of channel levees and beds before and after avulsions. Such investigations have been employed for
abandoned channels in deltaic settings (e.g. Carlson et al., 2020), and similar work could reveal the channel-scale mechanics
of abandoned channel attraction and repulsion in natural fluvial settings. Longitudinal studies of this nature could also
understand the rate at which abandoned channels are healed (and thus no longer affect pathfinding) and the direction or mode
in which they are healed, which we found to have important implications on avulsion dynamics (including lobe switching) and
long-term planform morphology (Fig. 12). If abandoned channel healing rates are observed to vary spatially (for instance with
distance along-strike from the active channel or distance along-dip from the mountain-front), this could motivate further
modeling efforts. It may be that healing proceeds in different directions and at different rates in different settings in the basin,
which will have important impacts on the spatial variation of avulsion dynamics and planform morphologies. We note that
detailed work on the time-fate of topographic highs associated with abandoned channels is especially lacking in the body of
literature. Finally, observations of avulsions in progress would help with understanding the appropriateness of our parameters
$\alpha_R$ and $\alpha_A$.

**6.4 Next steps & predictions for comparison with field sites**

We make several predictions that can motivate future observational and field studies. To begin, one key prediction is
that in the proximal portions of foreland basins, avulsions should be most-frequent on the surfaces of megafans (e.g., Fig. 6B).
These results compare favorably to the limited data available (Valenza et al., 2020) and can be tested by future observations
of avulsions in the available and future remote sensing record. The emergence of future datasets on real-world avulsions should
be able to confirm or deny the predicted abrupt, non-linear change in relative avulsion frequency with increasing distance from
mountain-fronts on megafans (Fig. 6B). These data about the location of avulsions should also allow testing of other predictions
from our model, including that avulsion in the distal domain of aggradational settings is more common immediately
downstream of abandoned channel confluences due to a greater total occupancy duration and therefore greater total aggradation
than either parent pathway immediately upstream (Fig. 8). Finally, our model suggests that stratigraphic systems with evidence
for clustering of channel avulsions (e.g. the Ferris Formation; Hajek et al., 2010) may have greater degrees of abandoned
channel influence via attraction or lobe switching than systems that appear more randomly distributed (e.g., the Williams Fork
Formation; Chamberlin et al., 2016).

**7. Conclusion**

Abandoned channels are pervasive on megafans in modern foreland basin settings. These locations also have some
of the highest avulsion rates in the observational record, which necessitates considering the role of abandoned channels on
avulsion dynamics and planform evolution in modeling efforts. We developed and presented a model that tests the interaction
between abandoned channels and an avulsing river. Our model intrinsically generates two distinct domains, proximal and
distal, in good comparison with remote sensing and previous research. We demonstrated that abandoned channels may shortcut
avulsion superelevation timescales in these settings by providing topographic lows adjacent to potential avulsion loci, by
providing remnant superelevation that can be inherited by future captured avulsions, including downstream of abandoned
channel confluences, and by transient knickpoint propagation that allows superelevated rivers to remain superelevated
upstream of the initial avulsion. The upshot of these factors is that avulsions are proportionately much more common over the
proximal distributive domain compared to the distal tributive one. We showed that tuning the degree to which abandoned
channel repulsion and attraction occur in simulations causes predictable changes in avulsion location during those runs,
whereby increasing repulsion pushes avulsions farther from the mountain-front, and increasing attraction pulls them closer.

Next, we demonstrated the important role that abandoned channel healing mode has on gross planform morphology, particularly over deep time, and that the proximal domain should grow until filling all available space in systems that heal via deposition-only or erosion-only. Finally, we have highlighted opportunities for future work by field workers and remote sensors in understanding the role that floodplain topography plays on avulsion dynamics, and the fate of floodplain abandoned channel topography.

## 8. Code Availability

Our model code is written in MATLAB and is publicly and freely available (under the GPL v3 license) via GitHub at the following DOI link: https://doi.org/10.5281/zenodo.5576789. The reference is included in our references list, under harrison-martin, 2021. This can be updated as needed during the review process.

## 9. Video supplement

A video supplement (Supplemental Videos 1-3) is uploaded to the AV Portal of TIB Hannover under the CC BY-NC-SA 3.0 DE license. The videos can be accessed at the following DOI links:

https://doi.org/10.5446/54887 - Martin and Edmonds Avulsion Model Supplemental Video 1

https://doi.org/10.5446/54888 - Martin and Edmonds Avulsion Model Supplemental Video 2

https://doi.org/10.5446/54889 - Martin and Edmonds Avulsion Model Supplemental Video 3

## 10. Author contribution

HM and DE conceptualized and designed the research and developed the code. HM collected and analyzed the data, wrote the manuscript, and prepared the figures. DE supervised the research and reviewed and edited the manuscript.

## 11. Competing interests

The authors declare that they have no conflicts of interest.

## 12. Acknowledgements

HM was supported by NASA FINESST grant 80NSSC21K1598. HM and DE were supported by U.S. National Science Foundation grant EAR-1911321. We would like to thank Ben Peters for assistance with preparation of Figure 1. We would also like to thank Gary Weissmann and Jeffery Valenza for helpful conversations about rivers in foreland basins.

| Symbol | Name | Units |
|---|---|---|
| $A$ | non-dimensional constant | non-dimensional |
| $A_{chan}$ | in-channel aggradation rate at some location | meters per year |
| $A_{fp,base}$ | base rate component of overbank aggradation | meters per year |
| $A_{fp,tot}$ | total overbank aggradation rate on the floodplain | meters per year |
| $\alpha_A$ | attraction factor | non-dimensional |
| $\alpha_{H,high}$ | healing rate parameter for high elevations | non-dimensional |
| $\alpha_{H,low}$ | healing rate parameter for low elevations | non-dimensional |
| $\alpha_R$ | repulsion factor | non-dimensional |
| $\beta$ | channel depth fraction | non-dimensional |
| $c_f$ | drag coefficient | non-dimensional |
| $C_0$ | bed sediment concentration | non-dimensional |
| $\eta_{aban,high}$ | abandoned channel levee elevation | meters |
| $\eta_{aban,low}$ | abandoned channel bed elevation | meters |
| $\eta_{adj,low}$ | channel bed elevation for a cell adjacent to the active channel | meters |
| $\eta_{appr,low}$ | low elevation in the cell from which a pathfinding avulsion approaches an abandoned channel | meters |
| $\eta_{chan,high}$ | levee elevation for an active channel cell | meters |
| $\eta_{chan,low}$ | channel bed elevation for an active channel cell | meters |
| $\eta_{farfield}$ | elevation of a far-field floodplain cell that has never been visited by the active channel | meters |
| $\eta_{fp,high}$ | floodplain elevation, high | meters |
| $\eta_{fp,low}$ | floodplain elevation, high | meters |

| | | |
|---|---|---|
| $\eta_{high,max}$ | the highest active, abandoned, or floodplain high elevation in a given row | meters |
| $h_{aban}$ | remnant depth of an abandoned channel | meters |
| $h_{avul}$ | flow depth of the pathfinding avulsion | meters |
| $h_{chan}$ | active channel depth | meters |
| $h_T$ | characteristic time needed to heal one mean channel depth | years |
| $\bar{h}$ | mean flow depth | meters |
| $H_{low}$ | healing rate for low elevations | meters per year |
| $H_{high}$ | healing rate for high elevations | meters per year |
| $L_h$ | levee height above approaching floodplain | meters per year |
| $q$ | normalized water discharge per unit basin width | square meters per year |
| $\rho_{sediment}$ | density of sediment | kilograms per cubic meter |
| $\rho_{water}$ | density of water | kilograms per cubic meter |
| $S$ | sediment specific gravity | non-dimensional |
| $\sigma$ | subsidence rate | meters per year |
| $t$ | time | years |
| $T_A$ | time needed to achieve superelevation | years |
| $\nu$ | diffusivity | square meters per year |
| $x$ | space | meters |

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
