# Peer review of "The push and pull of abandoned channels: How floodplain processes and healing affect avulsion dynamics and alluvial landscape evolution in foreland basins"

_Earth Surface Dynamics, 2021_

## Referee Comment (RC3)

**A review of:**
**The push and pull of abandoned channels: How floodplain processes and healing affect avulsion dynamics and alluvial landscape evolution in foreland basins**

Eric A. Barefoot

January 31, 2022

**Synopsis**

In this manuscript, Martin and Edmonds present a numerical approach to studying how abandoned channels affect flow routing and channel stacking patterns in alluvial fans. The authors formulate a random-walk model that finds a route for water and sediment on a fan surface, thereafter evolves the channel bed in one dimension along this path until an avulsion criterion is met, and then re-routes the flow according to a random walk until a new path is forged. This relatively simple algorithm is decorated with a few extra rules, which turn out to make a great deal of difference in the outcomes.

The first, and most important rule, is that previous channels in the landscape can either be a preferred path for the random walk, or an unpreferred path for the random walk when the algorithm is in the route-finding phase. This attractive or repulsive quality of the abandoned channels is a continuous variable for each that modifies the probability of a given random walk cell. The second rule is that channels do not persist on the landscape forever, and are "healed" according to one of three procedures: (1) eroding high elevations until only swales remain, (2) filling depressions until only ridges remain, or (3) the topography both raises and lowers until no topographic features remain.

Their model design is motivated by observations from large alluvial fans, where the authors see a large density of relict alluvial ridges, as well as a lower density of channels beyond some critical distance from the mountain front. The authors find that with these two continuous variables, and three mechanisms for abandoned channel modification, they can produce a rich diversity of outcomes in the model. In particular, they find that only the third (3) mode of channel healing is capable for achieving a steady-state fan, and that the degree that channels either repulse or attract reoccupation fundamentally shifts where avulsions occur in a strike-average sense. Moreover, their model results broadly mimic the general topographic features of the fans they drew inspiration from, lending some credence to the approach.

**Overall Comments**

I found this manuscript to be clear, well-structured, and very detailed. The model design makes a lot of sense, and I think the authors have shown a few very intuitive outcomes while also demonstrating a few less intuitive ones that spark interest. In particular, I thought the outcome where avulsion locations shift basinward when abandoned channels are barriers to flow was very intuitive, and makes for a satisfying result. In contrast, I found it surprising that imposing a rule that only negative or positive relief can be erased can drive the model to never achieve steady-state. These outcomes are presented and framed well, the conclusions are well-supported and impactful. My constructive comments are limited to a few minor comments on the visual presentation of the figures, and a few clarification questions on a few modeling choices. Other than these, I recommend the article be published. I look forward to citing this paper when my future work involves the stratigraphic architecture of fans.

**Minor Comments**

1. I have a question about this modeling choice. I am not sure if I understand why the simulation has to abort if a timestep results in a failed routing. If this were a real fan, the avulsion does not get a do-over, it has to fill the pond until it overflows, and then carries on its way. I wonder if by imposing this rule, you've introduced an artificial artifact of channel choice, where avulsions from the far-distant past can prohibit the present channel from traversing an entire sector of the fan. What if you adopted a really simple flooding algorithm instead? If while doing the random walk, the river encounters a dead end, it floods the area until it finds the nearest low point, and then starts routing from there.    line 325

2. I think you mean Equation 12 instead of 17?    line 341

**Figures**

– Throughout, I found myself struggling with the choices of colorbar used here. The authors are using `parula`, I think, which is a marked improvement in Matlab to the previous default colorbar, `jet`. However, `parula` is still not perceptually uniform. If a sequence of numbers that was strictly linear was plotted in `parula`, a viewer would perceive nonlinear jumps in intensity along the gradient. Put another way, there are features in the colormap that show up in plots that are not features of the dataset. To plot elevations, maybe try a single-hue colormap, or `winter` which I *think* is perceptually uniform.

– In general, I have one piece of feedback that applies to all your figures. As a point of style, you seem to have opted to putting a heavy black frame around every plot. While in the design phase, I can imagine it is helpful to have such a frame to see spacing between elements. For a finished product though, it intrudes on the visual space and commands attention, subconciously distracting the reader from the contents of the figure. Rules, used judiciously, can establish visual hierarchy (Figure 1 is a good example), but in a lot of cases here it is just too much. For all of your figures, I recommend getting rid of the bounding boxes.

– For example, here in Figure 2, the boxes around the annotations are essential, because otherwise the reader will never see them. However, I would remove the box around the figure, and take away the boxes around each of your colorbars. For these elements, proximity is all you need to establish a connection. On the subject of color, I would recommend a different colormap. The one here is distorting the visual presentation of the data. This colormap is really good at highlighting contrast in certain parts of its spectrum (e.g. yellow-to-red), and so the nice contrast showing the ridges that you want to see is limited to an arcuate band halfway through each map. I might instead recommend making four maps. In one pair, show just elevation in a single-hue colormap from light to dark so that the reader can see the conical shape. In the other pair, compute the slope map and plot that in a different single-hue colorbar. That way we can see both the ridges and the overall shape, but separated into two panels.

– Figure 3 is very nice, and seems intuitive and helpful, but it appears to have lost its caption.

– I like Figure 4 a lot. It's very helpful.

– In Figure 7, why do you think there is this odd, smoothly-sinusoidal lobe-switching? I didn't see it discussed in detail, but this is shockingly regular, and only seems to occur in the deposition-only or erosion-only healing modes

**Tables**

Usually for a manuscript, table design is not super important, but for ESurfD, it seems that they simply publish tables as-is, instead of reformatting them. Since this is the case, I have a few constructive comments that will make your tables much more legible.

– Vertical rules are not a great way of guiding a reader's eye. Alignment is a much better tool in the vertical direction.

– Horizontal rules are great for breaking up your tables visually into topical or related sections and for connecting items across rows, which is much harder for human eyes on the written page.

– So for Table 1, remove the box around the outside, and leave just the horizontal rule separating the heading from the table elements. See if you can make the individual cells single-spaced, while leaving some breathing room between cells. All the same for Table 2.

– For Table 3, do the same things like tightening the spacing within cells while leaving space between cells, also consider making the headings for each column bold. I would put the first column for figure references at the end. Also, the bit about having a parameter marked as "variable, see figure X" is very confusing, and forces your reader to flip back and forth. Instead, I would have groups of rows (set apart with horizontal rules) where you show every model run with every parameter combination. I know its a lot, but this table is already almost a page, so why not just make it a well-designed full-page table and go for it?

---

## Author Comment (AC1)

The manuscript submitted by Martin and Edmonds investigates the planform differences in megafans in foreland basins as a result of abandoned channels topography on avulsion pathfinding and their healing mechanisms. The study is motivated by three megafans and their channel locations and topographic features, including abandoned channels, alluvial ridges, and internal basins. A coupled 1D diffusive channel bed elevation model and 2D cell model is used to investigate both the effects of abandoned channels as attractors or repellents for avulsion pathfinding and the effects of different topographic annealing styles on modifying abandoned channel topography. Martin and Edmonds' results are very exciting and show that abandoned channels when acting as both attractors and repellents, create the characteristics topography of foreland basin megafans. Planform characteristics are also distinct between the proximal (near apex) and distal fan, which are directly related to avulsion potential and abandoned channel spacing. In their model, this spatial change in planform characteristics is only achieved if abandoned channel topography is transiently preserved and both high and low topography is diffused away over time. The megafan indefinitely progrades downstream if only the high or low topography of the abandoned channel is removed. These results provide a new and remarkable context to study the annealing of abandoned channel topography.

Below are major and minor comments and questions for the authors to consider.

- We appreciate the thoughtful and detailed review by Anonymous Referee #1. Two main themes in the review were to better support the motivation of the research by introducing a review of previous knowledge and modeling efforts earlier in the manuscript, and better supporting some modeling choices, including the decision not to vary floodplain overbank deposition rates with respect to distance from the active channel. We have changed the overall layout of the manuscript to address the first point, and we have added additional supporting justification for our modeling choices to address the second. We have also simplified some equations and added a new section on model parameter sensitivity. These suggestions, as well as all others in the review, are discussed per-item below. Overall, the reviewer is thanked for helping to improve the understandability and justifications of the manuscript and model.
- Note: line numbers referenced by the reviewer refer to the original manuscript, while (unless otherwise explicitly stated) line numbers in our responses refer to the revised manuscript with tracked changes accepted.

**Major Comments**

The introduction is a great summary of the motivation for this study. There is an opportunity to put the subsequent modeling efforts in context to other avulsion modeling frameworks, including those mentioned in the discussion and avulsion models related to deltas.

Besides avulsion processes, floodplain processes are the other addition to the modeling. The introduction glances over floodplain processes because not as much is known. Even with the limited knowledge about aggradation rates and their spatial trends on floodplains, the choice of

spatially varying floodplain aggradation rates away from the fan apex and not away from the active channel needs to be better supported.

Lines 81-82: How does this result compare to simulations?

- Of comparable simulations (i.e., preserves abandoned channels but only routes flow through one pathway at a time), it appears that Jerolmack and Paola (2007) and Reitz et al. (2010) have results that are similar to our proximal zone (densely channelized and distributive). However, they lack the tributive distal zone. We added a line to the start of section 3.2 to make this connection (172-173).

Lines 138-139: Currently, only the avulsion location variation is plotted. What are the implications for timing?

- Avulsions become more frequent when relative superelevation (Eq. 2) is considered, compared to only the original formulation (Eq. 1). This is shown in Figure 6B. We have made this more explicit by adding the average time between avulsions to the caption for Figure 6, and to lines 519-520

Lines 213-214: Please clarify why only non-active channel cells have subsidence, especially since subsidence is accounted for in the equations of both the aggradation rates and 1D diffusive channel bed elevation model.

- Thank you for pointing out this lack of clarity. In the model, active channel cells do experience subsidence. However, it is implemented elsewhere in the routine (at the same time as the channel elevations are updated) separately from floodplain cells for logistical reasons that do not affect the outcome. As such, I have adjusted the text (260-261) to remove this unnecessary detail and make it clearer for the reader.

Lines 214-215: Please describe the motivation for varying floodplain aggradation away from the mountain front but not the active channel. Does this affect the planform result and avulsion patterns? What are the processes that distribute sediment across the floodplain in this system, especially with only one active channel (Line 84-85)?

- Our reply regarding not varying floodplain aggradation with lateral distance from the active channel is provided in the reply to the comment on Lines 331-332. Regarding floodplain aggradation variation with respect to the mountain-front, our intuitive reasoning was that overbank deposition rate should increase with distance from the mountain-front as grain size fines and washload concentration increases. With that said, later testing showed that we still reproduced the fundamental findings of the manuscript with a base deposition rate that does not change with distance from the mountain-front. A sentence explaining as much has been added to our results (478-479) and to a new parameter sensitivity section, 6.1 (656-660).
- In the absence of spatial variations, the role of floodplain aggradation is only to counter-act subsidence. This difference is most pronounced in the proximal region where subsidence is greatest and pathfinding greatly restricts the ability for the active channel to

visit regions that are lateral to the origin and very proximal to the mountain-front. In the real world, mountain-front shedding processes or other channels should fill these topographic lows, but it is an unavoidable consequence of our decision to "ignore the impacts of other rivers or fans and of any other mountain-front processes that may advect sediment into the basin" (Lines 230-231). We have put a mention of this phenomenon in section 6.1, 654-656. Instead, our simplified model addresses this proximal "sag" by employing overbank aggradation that consists of a variable rate that increases as topographic relief (between each row's highest levee and far-field floodplain elevation) increases. This is a somewhat simplified version of Jerolmack and Paola (2007)'s model implementation (Equation 3 at https://doi.org/10.1016/j.geomorph.2007.04.022). We do not believe that Jerolmack and Paola (2007)'s justification for this implementation is entirely unreasonable, as regions that are inundated to greater degrees should experience more aggradation. We do agree, however, that this could be presented more clearly and explicitly to the reader, and we have reworked this overbank sedimentation section of the manuscript to address any readers' concerns that the reviewer helpfully pointed out. 383-400.

Lines 331-332: See previous comments. The motivation for floodplain aggradation changing downstream but not with distance to active channel seems counter-intuitive. Please explain how results would be affected if floodplain aggradation rates varied with distance to channel.

- By design, our model treats channels as sub grid-scale features that are small relative to the size of a cell, such that "channel-scale processes (like meandering, crevasse splays, or other lateral-distance-dependent depositional effects) are not resolved" (lines 289-290). Overbank deposition is generally modeled as decaying exponentially with increasing lateral distance to the channel (ex. Pizzuto 1987, at https://doi.org/10.1111/j.1365-3091.1987.tb00779.x0) and for a given suspended grain size (very fine sand) this should decay to a very small value within one or two cells: assuming quartz grains in water, application of Stokes' law yields a settling velocity of ~0.005 m/s for very fine sand (75 um) and ~0.0005 m/s for silt (25 um). If we assume the flood is 1 m deep and there is an aggressive overbank velocity of 0.5 m/s, then very fine sand and silt would travel ~100 m and ~1000 m, respectively, before settling. The same exercise with clay-sized particles yields transport distances >10 km, which represents our overall floodplain deposition term. In other words, the sedimentation away from the active channel is subgrid-scale for the suspended load. If floodplain aggradation rates did vary with distance to the channel (beyond some small distance that formed local levees), we would observe broad, continuous, and uniform alluvial ridges which we do not see in our remote sensing observations (Figure 2). With that said, we think that the results could be affected in interesting ways if abandoned channel healing (via preferential deposition in topographic lows or erosion of topographic highs by overbank floods, specifically) were to vary in rates with lateral distance from the channel. We further clarified our sub grid-scale design in our floodplain processes methods. A sentence explaining this as a potential direction for future reference has been added to the discussion (6.3). Lines 708-711.

Line 335: A_fp,f is listed in Table 2 as having two different rates for proximal and distal. Please describe this variation here and share how the boundary between proximal and distal is found.

Does the boundary location change for each timestep? And how would these results differ if only one fix rate existed?

-   We appreciate the reviewer identifying a possibly confusing issue for readers. As mentioned earlier in our response to the comments on lines 214-215, we intended for overbank deposition to increase downstream reflecting a supposed increase in washload concentration away from the mountain-front, with the value for each row being linearly interpolated between two end-member values. We explained this sort of interpolation between two end-member values for subsidence in our manuscript, but neglected to do the same for overbank aggradation. In other words, for the results presented in this paper, we did not establish a boundary between proximal and distal for the purposes of overbank aggradation but instead applied a linearly increasing amount per row. As mentioned in the previous reply, testing with a base rate that does not vary with distance from the mountain-front reproduced the fundamental findings of the manuscript, including reproducing the two distinct zones. This has been added to lines 656-660.

Lines 335-336: Please consider presenting the formula for $A_{fp,v}$ here. Based on the description given, it's not clear if this is a positive or negative linear relationship with height difference.

-   In response to the comment and in an effort to simplify our overbank deposition description, we have removed the fixed component from our Equation (12) and re-ran the models. This did not have a significant effect on our model outcomes. We have adjusted the explanation and formula to reflect this change, and also to make it clearer that our overbank deposition rate is equal to a base rate times a scaling factor that increases with increasing relief between each row's highest levee and far-field floodplain elevation. We also ensured that we stated the directionality of this linear relationship. Line 383-400.

337-338: Does channel depth also change downstream? What are the motivations for normalizing the floodplain aggradation based on channel depth?

-   It does, but the $h_{bar}$ term in this equation (as defined above for Eq. 6b) refers to mean channel depth as averaged over the whole active channel, and so does not change with distance from the mountain-front. We added text to remind readers of this here (391). Our reasoning for normalizing the elevation difference between each row's highest levee and farfield floodplain by channel depth is to create a non-dimensional scaling factor that can be applied to a base aggradation rate; it is necessary to divide this elevation difference by some value with units of meters in order to ensure dimensional homogeneity on both sides of the equation. This is the same normalization employed by Jerolmack and Paola (2007) to ensure dimensional homogeneity, and a citation has been added to reflect that (386). Our equation and text has been changed (383-400) to hopefully make this clearer, and we especially think that changing $A_{fp,f}$ and $A_{fp,v}$ to the single value $A_{fp,base}$ should improve this section's clarity.

Equation 12: See previous notes (and please feel free to refer to your answers there). The model presented here is a physical-based cellular model (line 13). Please describe how this formula is inspired by our communities' understanding of floodplain aggradation processes.

- While our implementation is certainly simplified, we believe that it is not unreasonable. The updated formula employed in this revision now states that overbank aggradation increases linearly with increasing relief between the highest levee (often the active channel) and the floodplain. We added text (389-390) stating that "[w]hile simple, this scaling reflects a basic intuition that regions of the basin that are inundated to a greater depth during flooding should receive more overbank sediment." This is the same justification employed by Jerolmack and Paola (2007), but we certainly agree that this is likely a rich problem that we treat crudely, and perhaps future experiments that are more sensitive to implementation choices could be opportunities for further research.

Figure 4: In the physical-based model, how would the deposition only or erosion only healing modes affect the sediment availability in the surrounding fields since either sediment is needed or transported away in these modes. Would adding a component accounting for this in the floodplain aggradation affect the potential for equilibrium to be reached (lines 20-22)?

- The reviewer is correct to point out that a more sophisticated model would account for the mass transport from surrounding cells toward in-filling abandoned channel bases, or toward surrounding cells for abandoned channel levees. While one could perhaps argue that the sediment removed via erosion will be transported as washload and thus leave the domain without conserving mass (ex. Jerolmack and Paola (2007) in justifying channel incision rules), our motivation for not simulating these effects was ultimately due to computational efficiency; solving a 2D sediment transport equation over the whole domain once per timestep would add prohibitive time per run. We are interested in pursuing this idea for future research, however. For the purposes of this manuscript, we have updated section 5.5 to add some context about how sediment availability differs between the three modes. The reviewer does identify a key difference between the three runs, which is that they result in different amounts of floodplain mass over time, which will affect the final topography (as is evident in Figure 12) and lobe switching. Our new explanation of lobe switching (section 5.5) reflects this different in accumulated mass/topography on the floodplain, as well.

Line 357: Please be more specific of the location in the discussion, where the choice of 55,000yrs as a healing timescale is explained. How does the healing rate compare to floodplain deposition rates described previously?

- The reviewer correctly identified a mistake whereby the newest version of the manuscript did not have a matching section in the Discussion. Instead, in this revision, we have added a justification and discussion around the choice of healing timescale directly to this section. Thank you. Lines 411-424.

Section 4.1. The planform topography and feature similarity between observations and the model are striking. A short discussion on how channels bounding some of the observational fan (Fig 1 E, F) could be affecting the along-strike comparison would be helpful for context. Additionally, a short description of the 1D model validation, including the water depth of the channel, where available, would be a powerful addition to this section, especially since mean water depth dictates the healing rates (Eq 13).

- We added a mention of the lack of external topographic controls (including bounding rivers) to the end of section 5.1 (491-492). Additionally, we added a short 1D model validation component to section 5.1, including a mention of reproducing appropriate channel depths and slopes (471-473). We elected to not include a more sophisticated validation section due to the emphasis on reduced complexity modeling as opposed to reproducing all details in the environment. We believe that we have reproduced the essential topographic features and fluvial geometries necessary to investigate the roles of abandoned channels in shaping these mountain-front environments in a general sense.

Line 411: I would encourage acknowledging that some parameters were varied between the proximal and distal zone (Table 1). Therefore, it's unclear if the results are affected by the choices in parameter variations.

- The reviewer is correct to state that certain external parameters did vary with distance to the mountain-front. Specifically, these were the overbank aggradation base rate and the subsidence rate. We added this acknowledgement to the relevant line (478). Additional modeling efforts have shown that we reproduce the same essential findings of the paper even within a basin that is subsiding uniformly or without a base aggradation rate that changes with distance from the mountain-front. A sentence reflecting these findings has been added here (478-479) and to a new section 6.1 discussing model sensitivity to non-experimental parameters (656-660).

Figure 6: Accounting for surrounding topography in $T_a$ (Eq 1 and 2) shows a striking increase in avulsion location, especially in the proximal part of the fan. How different are the corresponding $T_a$ distribution (Eq 1 and 2) between the red and blue line model runs?

- Because the red line (Eq. 1) in Fig 6B is normalized to the same maximum value as the blue line (Eq. 2), the red line shows that the time between avulsions is about twice as long in the red line. I have added the average time between avulsions for each case to the caption and also to lines 519-520.

Figure 7: Are the striking differences between distal and proximal related to slower healing rates and different water depth ($h_{avuls}$ etc) for avulsion pathfinding?

- Abandoned channel healing rate is held constant spatially and temporally within each run, and therefore does not vary between proximal and distal domains. With our new runs that do not employ changes in base overbank deposition rates with distance from the mountain-front, we still reproduce the findings of the manuscript. With regards to channel depth, the reviewer is correct that channel depth does vary spatially because "depth [varies] as a function of local slope" (section 4.2). As slope is generally steeper on the proximal fan surface than in the distal domain, channels are generally somewhat shallower in the proximal domain than the distal (~1.2m vs ~1.8m for an example run). We hypothesize that this change in channel depth is a result of the emergence of the two domains (since one is steeper than the other), as opposed to the opposite causality. To test this hypothesis, we performed additional runs where we replaced the $h_{avul}$ term with $h_{bar}$, or global mean channel depth, which does not vary spatially. Even in this case, we

still generated the two distinct domains. A sentence reflecting this has been added to the new section 6.1 (656-660).

Before describing the main discussion points, a short overview of the sensitivity of the results to slight variations in floodplain aggradation and subsidence rate (Table 1) and the ratio between them would be beneficial in a section.

- Done. Added section 6.1.

Line 648-649: One set of studies that have been conducted related to channel beds and levees during and after avulsions has been led by Dr. Brandee Carlson on the Huanghe River, China. Please consider including the findings of the studies here.

- Thank you for bringing this work to our attention. While the mechanisms are not identical (as Dr. Carlson et al.'s work on the Qingshuigou Lobe primarily employs marine/tidal sediment), we can at least make reference to analogous work exploring abandoned channel annealing mechanics in another depositional setting. Added reference (703-705).

**Minor**

Line 10: Previous modeling work is portrayed as if they are vastly different from this study. I am not sure that's representative.

- We hope that the revisions to the manuscript that resulted in the addition of a background information section (2) early in the document will address this concern. This section explicitly frames this study as filling a research gap in the work of previous models, namely, the conscious inclusion and testing of abandoned channel dynamics.

Figure 3 is missing its caption.

- Correct. This has been fixed; thank you!

Section 3.2: Does the choice of the 1D long profile elevation model affect the profile, water depths, and sediment transport rates?

- Our 1D long profile elevation model is rather simple (diffusion) other than for the ability to transiently transport sediment instead of immediately coming to equilibrium between timesteps. This transient nature is important to include as it has effects on avulsion dynamics (Fig. 8; Fig. 9). Implementing a more sophisticated sediment transport model may refine our channel geometries, but we haven't noticed a particular sensitivity to water depths or sediment transport rates for our key findings.

It's unclear how the 1D model is used to initiate the 2D model. I would suggest including a plot of the initial 2D domain.

- Due to the length of the manuscript as-is, we have elected to add additional clarifying text to the relevant section of the methods (238-241) as opposed to adding a novel figure. We hope that this serves to clarify any confusion for readers that may have otherwise arisen.

Line 148-150: This information would be extremely helpful in the introduction.

- We agree, and have added the background information section (2) to address this.

Table 1: Please add notes to the caption that describe the motivation for the choice of parameters value. Apex elevation is missing in table 3.

- The specific values are not critical to the operation of our model nor our findings; we now discuss the sensitivity briefly in section 6.1. Nonetheless, we have added text (242-243, 471-472) explaining that our intention was not to simulate any specific river on earth, but rather to choose reasonable values to recreate reasonably representative rivers and megafans in a broad sense. This is in line with the philosophy of reduced complexity modeling. Apex elevation should have read "initialization length"; thank you for the catch.

Line 201: Will subsidence affect surface slope in the model?

- Yes, but not to an appreciable degree within reasonable ranges, unless it is significantly greater than aggradation. In that scenario, such effects can arise over millions of years of simulation time. This was added to section 6.1 (654-656). Otherwise, aggradation due to the channel is the principal determinant of surface slope on the fan.

Table 2: Please add notes to the caption for the motivation of the parameters. Especially sediment discharge, incoming sediment supply, and basin width.

- We hope that the newly updated caption on Table 1 will service as a disclaimer for readers, and does not require duplication for the caption on Table 2. (See response to "Table 1" above).

Line 232: It's not clear how Table 2 relates to solving h_chan. It would be interesting to include a description of how h_chan is found.

- While we agree that such an inclusion would be interesting, in the interest of length, we decided not to include an explicit step-by-step but instead to refer readers (line 278-279) to Paola et al. (1992), whose methodology we employ. Readers interested in our specific code implementation may find interest in reviewing our publicly available (and commented) code.

Line 262: Please include where the formula for healing rate is found in the manuscript here.

- We have ensured that this section refers to section 4.2.2, which contains Equation 13.

Lines 268-270: I am confused by this statement. Could floodplain aggradation allow for the elevations of these cells to increase?

- Floodplain aggradation can increase the elevation of these cells, but cannot increase the relief between abandoned channel highs and lows.

Lines 289-290: Please clarify to which cells this applies Are floodplain and abandoned channel cells equally likely to be a site for avulsion triggers?

- Clarified (336-338) that we calculate this over the first step into surrounding cells (agnostic of the identity of what that first step could be, provided it is not the parent river from which the avulsion originates).

298-299: Please specify why 30 years was selected here. How does this compare to results from equation 2 for model runs?

- Added a brief explanation for 30 years (346-347). Added a mention of the mean time between avulsions for Eq (1) and (2) to the caption/figure of Fig 6, and to the text in lines 519-520.

Line 310: Please describe how is h_avul is calculated.

- We describe how h_avul is calculated on the lines (362-363) immediately following the equation. Essentially, we assume the avulsing flow is channelized, and calculate its depth as we would an active channel cell.

Line 341: I think this should read equation 12.

- Correct, fixed.

Figure 5, 6, 10, etc. A more complete description in the caption of the planform elevation model results would be useful, including a color bar scale and a description of what is plotted – I am assuming it's all cells that have been channels at one point in the modeling.

- We mention that the output is detrended high elevations. The colorbar does, in fact, have a scale. We have added elaboration on the colorbar into the caption, however (498-499).

Lines 433-434: Please clarify this statement. Currently, (b) doesn't appear to have a vertical scale.

- By "vertical axis scale for (b) is the same as (a)", we meant to say that the y-axis on panel b uses the same tickmarks and units as panel a, i.e., distance from the mountain-front in km. We have added a duplicate label to Figure 6b to avoid confusion.

Lines 507-510: This downstream shift in sediment conveyance because of internally drained floodplains is an exciting result and makes a lot of sense physically.

- We agree, and it's nice to have some intuitive results come out of the model!

Lines 535-537: I wonder if one of the implications for this result is that more sediment will exit the domain at the downstream extent. Does this make sense with the physical-based framework set up here?

- We would agree with that implication. We believe that this does make physical sense. A river at equilibrium would have an amount of sediment extracted from it that exactly keeps up with subsidence. A river flowing into an underfed basin may, instead, have all of its sediment extracted from it before exiting the basin. As the number of possible pathways increases, the same quantity of sediment is being "spread over" multiple pathways, keeping them further from equilibrium and sequestering more sediment proximally, as evidenced by further propagation of superelevation resulting in the avulsions further into the basin seen in these more repulsive runs.

Lines 581-592: This paragraph would be a great context for the introduction.

- Agreed, and moved to the background information section (2).

Lines 596-597: This is super interesting, especially since it's in agreement with previous findings.

- We agree! It's exciting to imagine that different pathfinding rules apply at different distances from the mountain-front.

Lines 611-625: This paragraph would be a great context to motivate the study and results.

- Agreed, and moved to the background information section (2).

Lines 649-650: The sentence is currently not complete.

- Fixed, thank you.

---

## Author Comment (AC2)

General Comments

The manuscript presents novel and interesting results from a new cellular model of river avulsions in a foreland basin setting that explicitly parameterizes the influence of abandoned channels in channel pathfinding. Although several models of river avulsion exist, there has been a lack of attention to floodplain topographic controls on flow routing, and this manuscript presents an interesting and logical new parameterization of the ways in which abandoned channels might control avulsion pathways. They treat abandoned channels in three ways: 1) their repulsion caused by the elevated topography of alluvial ridges; 2) their attraction because of channelized flow paths; 3) different modes of abandoned channel healing, depending on abandoned channel deposition and erosion. Model results show that all tested aspects of abandoned channels control avulsion location, especially whether avulsions are occurring in the proximal or distal reaches of the fan. They also show, most interestingly, that active lobes and lobe switching – that is, when avulsions occur in a clustered region of the fan before switching to a new, clustered region – only occurs under certain abandoned channel healing conditions. This suggests that abandoned channel topography exerts a strong control on the locus of avulsions and also the tendency for some fluvial fans to have active lobes. This is a valuable modeling contribution that sets the stage for field and remote sensing work to gather more real-world data about the role of abandoned channels to test some clearly defined model predictions. At the beginning of the manuscript, they also characterize the amount of abandoned channels on three fluvial megafans, to show that these are indeed prevalent features in modern environments. Overall, this is a well-written manuscript with exciting findings; the comments below are mostly minor questions or suggested revisions to improve the clarity of the manuscript before publication, although there are a few more significant comments, especially about the default channel healing mode used in most of the model runs.

- Dr. Chamberlin has provided a thorough and useful review. Common themes that arose in the review were i) a need to move more of the background or literature review information from later sections to an earlier part of the manuscript, ii) extensions to the discussion and greater interpretation of some results, particularly lobe switching, and iii) additional explanation or justification of some of the model design decisions. We have been able to adopt most of these changes, particularly through the introduction of our new section 2 that collects background information and motivation and moves it earlier in the manuscript. We also took the opportunity to replace the literature review that used to be in the discussion with a model sensitivity section as well as additional interpretation of the results and applications or predictions regarding the stratigraphic record. Finally, we added to our design justifications throughout. We appreciate the reviewer's careful attention and thoughts, as well as their kind words.
- Note: line numbers referenced by the reviewer refer to the original manuscript, while (unless otherwise explicitly stated) line numbers in our responses refer to the revised manuscript with tracked changes accepted.

Specific Comments

**Base run and validation**

In section 4.1, the authors validate the model results by comparing a baseline run to 2 modern megafans. However, it is unclear why the strike and dip-section profiles are compared to two different megafans. Is the dip-section of the model inconsistent with the Taquari fan? Is it important to have reasonable consistency in both dimensions? A lot of the main conclusions of this paper are about the proximal-distal location of the avulsions, so the length-to-width ratio of the fan area does seem like an important property. Also, for the along-strike comparison with the Taquari fan, the Y-axis values are comparable but the x-axis values are very different; the plots make them look the same, but the axis values are different. How important is this difference? Additional text should be added to address this, and ideally the dip and strike profiles should be compared from the same fan system or systems.

- The reviewer notes an important limitation in the ICESat-2 data we used for our validation. ICESAT-2 data are limited to ~north-south oriented tracks (see two following images). As a result, we are unable to measure both along-strike and along-dip orientations for any one individual fan, necessitating the use of two fans. Within any individual fan, the model can reasonably recreate the length-to-width ratio of fan areas. In searching for a fan by which we could compare multiple along-strike cross sections, we encountered several obstacles. Unfortunately, the spacing between transects in ICESat-2 is rather wide (order kilometers), meaning that only very large (i.e., greater than the width of the simulation in our model) fans can have multiple along-strike transects measured, and there are relatively few of these that are generally oriented east-west with a minimum of human landscape disturbance and entirely continuous data without data gaps due to cloud cover on the day of measurement. Recognizing this limitation, we had believed that a reader would still benefit from seeing that we "recreat[e] the change in along-strike profiles" as one moves downstream. As a sort of disclaimer, we mentioned that the reader should "[n]ote the exceptional vertical exaggeration in the Taquari along-strike profiles". With that said, we now recognize that this could confuse readers, and to avoid confusion would require adding a section about limitations in ICESat-2 data, which would distract from the point of the (already long) paper. As such, and because the content is not necessary for the core conclusions of our model, we have simplified the figure to only present the along-dip comparison to the Pucheveyem and updated the text accordingly in lines 467-476.

[Figure]

[Figure]

**Abandoned channel healing mode in the first 3 sets of model runs**

Lines 374-387 describe four series of model runs that are then analyzed in the results section. If I understand this correctly, series 1-3 were all run in far-field directed healing mode, and thus the effects of the repulsion and attraction parameters were tested only with the far-field healing. However, in the fourth series of model runs, the authors show that the depositional and erosional healing modes cause major (and interesting!) changes in avulsion behavior. Do the impacts of

the attraction and repulsion parameters still occur under the depositional/erosional channel healing modes, or are they only important variables under the far-field directed healing mode? It would be good to see some model runs added (perhaps just a limited subset of the model space) that address this question. Is the channel healing mode more important than the attraction/repulsion rules? Also, what is the justification for using the far-field directed mode as the default mode? Is there evidence to suggest that this is the most common/reasonable healing mode in modern floodplains? The authors mention that this is the least computationally intensive, but it would be good to see scientific justification for this as the default mode.

- The reviewer raises a great point in that our experiments on abandoned channel repulsion and attraction use the far-field directed healing mode, and we do not show corresponding results using the other two healing modes. The reasons for this are multiple. As a scientific justification, we began with far-field directed because it is the only healing mode that is truly able to completely "heal" topography, i.e., erase the topographic memory of both abandoned channel highs and lows after finite time. Of the three options, this is the closest to the endpoint of diffusion on an infinite plane. More practically, these results do hold for the other healing modes, but their effects are much more difficult to measure and display quantitatively as these two modes do not achieve dynamic equilibrium. As such, the effects these processes have on the location and abruptness of the transition between proximal and distal domains would have to be observed transiently during the first few Myr of simulation as progradation is ongoing. As such, for any specific distance from the mountain-front, measurements of spatial avulsion frequency necessarily integrate over both a distal and proximal phase of varying durations. It is also worth mentioning that all three modes are equally computationally intensive, but all three are far less computationally intensive that employing true 2D diffusion on the landscape at each timestep. This was our intended point of communication in our justification of far-field healing in section 4.4 (lines 444-446).

**Model mass balance variations based on healing mode**

Because of the way this model is set up, mass balance is not constrained between model runs, and that is not an impediment to the analysis. However, the different healing modes shown graphically in Figure 4 have very systematic differences in mass balance that might have big impacts on the model results. For example, the depositional only mode of healing would require a much larger sediment input to the floodplain than the far-field directed or erosion-only modes. There is no mention of this in section 4.5 or in the discussion, but I think this difference in mass balance between the model types warrants some analysis. If equivalent amounts of sediment were added into the proximal floodplain in the far-field directed healing model runs, even outside of the abandoned channels, wouldn't that also cause lobe switching? In other words, to what degree are these results caused by the abandoned channel healing versus just accumulation of more or less sediment in the floodplain around the active channel belt?

- This is a keen observation. We do believe that each mode confers a different amount of sediment and a different preservation potential for abandoned channel topography. This is likely inextricable from abandoned channel healing, considering how much of the floodplain is constituted by abandoned channel topography (Fig 2). We have elaborated

on the origin and nature of lobe switching in the results (614-627, 632-637), and added to our discussion (693-697) with content about what lobe switching may or may not say about abandoned channel healing.

**Re-organization of introduction & background**

The introduction section of this manuscript is very short (3 brief paragraphs!) and does not give enough background to set up the hypotheses or the model set-up. Other sections of the paper (specifically section 2.2.2 and Lines 611-625 of the discussion) would be better suited to this introduction, so that more specific background about abandoned channels and cellular modeling of avulsions was provided to the reader before they continue on in the paper. Also, the topic of avulsion reoccupation is not mentioned in the introduction, but the observations from the modern and the ancient that channels commonly reoccupy previous channel courses is critical to the motivation for this paper. There should be at least some description of the evidence for channel reoccupation of abandoned courses in the introduction here.

- On the one hand, we stylistically prefer a concise introduction, and felt that doing too-detailed of a literature review in this section would interfere with the "flow". Additionally, we felt the need to demonstrate that abandoned channels are prevalent on fluvial megafans before discussing why they should be considered in modeling efforts. On the other hand, multiple reviewers have made the same suggestion, and it is a reasonable one. In retrospect, we admit it's a bit odd to have that much of a paper's justification in the Discussion. As such, we've added a new section between the introduction and remote sensing sections (2) titled "previous modeling works" in which we motivate the future work. Thank you for the suggestion.

**Discussion section revisions**

As noted with detailed line numbers in the following section, a lot of the discussion reads more like background about avulsion models and justification of the model rules used in this paper, rather than contextualizing these novel results. I think the background information and justification of model parameters should be moved earlier in the manuscript, and a more detailed analysis of the results could be added to this discussion section – especially thinking about their broader implications. For example, based on the prevalence of lobe-switching in modern fans, does this suggest that the deposition/erosion-only healing rules are most consistent with modern observations? Additionally, what are the implications for stratigraphic analysis of avulsion patterns? In systems with clearly clustered avulsions (such as the Ferris Formation; see Hajek et al. (2010)), does that suggest abandoned channels were attracting avulsing flow more than in randomly distributed systems (e.g., the Williams Fork Formation; see Chamberlin et al. (2017))?

- We have moved much of the discussion to a new background information section (2). In its place, we have reworked and added additional content explaining and discussing predictions for how different avulsion emplacement "rules" could reflect different parts of either parameter space or positions on/off of a fan in our models (670-688), with a

particular mention of lobe switching (679-681). The predictive power of lobe switching on abandoned channel healing mode is now mentioned (693-696). We have included a brief reference to connect the work with the Ferris and Williams Fork formations to the end of section 6.4 (724-727) as suggested; thank you for the improvement.

Technical Comments

Section 2.2 - Megafan floodplain topography discussion: the organization/title of this section is odd, because this is really the background needed for the model set-up, not a discussion of the field results. I think almost all of the content in this section would be better in a background section about avulsion set-up, initiation and pathfinding that would come before the remote sensing section.

- We have added a background information section (2) that will hopefully help address concerns about the order in which model set-up is presented.

Figure 3: missing caption

- Correct. Fixed!

Line 140-141: There is also good evidence from the rock record that avulsion reoccupation of previous channel courses is common – e.g., see references in Chamberlin and Hajek (2015).

- Added (62-64, 212-214). Much appreciated.

Line 161: There are several studies that have observations of oxbow lake sedimentation rates, which is a type of abandoned channel sedimentation. These studies should be cited and discussed here. (for example, Wren et al., 2008, "The evolution of an oxbow lake in the Mississippi alluvial plain").

- Added, thank you. Lines 88-90, 413-415.

Line 163: The language "if one assumes that abandoned channels do heal" is confusing – what would another option be? Over geologic time, they must heal, right?

- I agree with you, but in conversations with colleagues, it seems that not everyone is convinced. In aggradational basins they believe that the healing timescale of abandoned channels may be very long relative to the burial timescale, meaning that in a functional sense abandoned channels would not heal.

Line 168: The details in the discussion should be moved up front into a background section.

- Done, added section 2.

Table 1: Maybe "numerical" or "cellular" model is more clear than "non-experimental" in the table title?

- Done (242).

Line 263: This is an interesting way to code channel healing! Can you add some more explanation – maybe here, maybe in the justification for the model set-up – about the mechanistic justification for these different healing modes? In other words, for the depositional only healing, how would that actually work? Via overbank sedimentation? Via temporary reoccupation of the abandoned channels during floods? I know that this model is not attempting to resolve those processes, but some general outline of how each of these healing mechanisms could be possible would be helpful for thinking about the implications.

- This information was previously located elsewhere in the manuscript (old 2.2.2). You are correct that it's difficult to resolve the processes, or even to evaluate which should be more common, which is why we treated it as an experimental parameter. With that said, it's certainly true that it'd be helpful for a reader to have some sort of physical mechanism in mind, so I've moved it to a more fitting place with other abandoned channel healing considerations in the new Previous Modeling Efforts section (2), and added some additional context.

Line 333: I'm surprised by this decision to have floodplain aggradation independent of active channel position. Why are they decoupled? Additional justification of this choice would be helpful, because this would have a big influence on superelevation dynamics. This might be a point that would be good to add into the discussion and a direction for future modeling work.

- This comment was similar to the comment provided by Anonymous Reviewer #1, and a detailed response can be found in our reply AC1 under the reply to comments on Lines 331-332. We reproduce it here in brief: we make the assumption that the diffusion distance perpendicular to the flow direction (Pizzuto 1987) is contained within a single cell. More specifically, we assume that the coarse grains of the suspended load (very fine sand & silt) are not transported beyond the grid cell during a flood, instead contributing to alluvial ridge or levee formation. Assuming quartz grains in water, application of Stokes' law yields a settling velocity of ~0.005 m/s for very fine sand (75 um) and ~0.0005 m/s for silt (25 um). If we assume the flood is 1 m deep and there is an aggressive overbank velocity of 0.5 m/s, then very fine sand and silt would travel ~100 m and ~1000 m, respectively, before settling. The same exercise with clay-sized particles yields transport distances >10 km, which supports our uniform component of overbank sedimentation representing the sum total of all processes that deliver sediment far from the active channel (e.g., washload in overbank flooding, pluvial flooding, overland flow).
- With that said, I think that there is likely some rich behavior to explore specifically regarding spatially variable abandoned channel healing. The rates, and maybe even modes, of healing could be easily justified to vary with distance from the channel (hydraulic connectivity, washload differential deposition in topographic lows, erosion/reactivation potential during overland floods), or with distance from the mountain-front (micro-climatic or vegetation differences, increasing washload downstream, etc). These explorations are beyond the scope of this study, but we have ensured that a brief discussion of these factors is included in section 6.3 (708-711).

Line 341: I think you mean equation 12, not 17

- Correct, thank you.

Line 347: Put the detailed explanation of other models in the background, not discussion, and then this line can be removed from this paragraph.

- Done.

Figure 4: this is a very helpful figure that I referred to many times when reading this!

- Thank you for the kind words!

Table 3: I think presenting model run parameters based on figure # is confusing. It would be more straightforward to include a table that shows the parameter ranges for each of the four series of model runs.

- Agreed. I've updated the table to remove reference to figures to 'see parameter ranges', and instead put ranges in the table itself (438). I've also redesigned the table entirely as per RC3's suggestions.

Lines 472-474: Interesting! Is there any evidence for this in modern systems, that avulsion nodes cluster immediately downstream of previous avulsion nodes? This would be an interesting point to expand on in the discussion.

- This is an interesting suggestion, but we know of no way to test this idea with modern data.

Figure 12: Make dashed lines have a bigger dash, they are hard to distinguish from solid lines. Also, typo in the first part of the caption – missing "on" before (a).

- Thank you for the catch; the dashing was supposed to be the same form as in Figures 10 and 11 but I missed it. Updated, as is the typo in the caption.

Line 592: Abandoned channels impact the compensation rule in the Chamberlin and Hajek models because they leave an elevated abandoned alluvial ridge, so they cause repulsion away from the abandoned channels in the compensational rules.

- Thanks for the clarification! Added to the sentence, which is now in section 2 (75-78), and clarified that it is moreso the role of upstream flow routing that isn't necessarily resolved.

Lines 596-598: The clustered mode in the Chamberlin and Hajek (2015) model randomly selects a channel location within the clustering zone; it only selects the lowest location when an elevation threshold above the far-field floodplain has been reached.

- Ahh, my misunderstanding. I removed the erroneous connection. Thank you.

Paragraph beginning at line 611: I think this would be better suited as background material.

- Done (see response to "Re-organization of introduction and background")

Lines 637-647: This material would be helpful for justifying the healing rules used, and thus could be presented earlier in the paper.

- Done (see response to "Re-organization of introduction and background")

Section 5.3: These are interesting predictions focusing on the proximal-distal location of avulsion nodes! I think adding predictions about channel healing & lobe-switching behavior would be very helpful too. Are there characteristic types of fans that show lobe switching? Is there something about the floodplain aggradation style on those fans that influences abandoned channel healing? That would be really cool to explore.
- We have added some additional analysis on lobe switching to section 5.5 (614-637), and additional discussion to section 6.3 (690-714). We agree that this is an exciting research direction to pursue in the future, and have tried to add predictions and guidance that could help precipitate or facilitate this work.

---

## Author Comment (AC3)

Reviewer #3: Eric A. Barefoot

**Synopsis**

In this manuscript, Martin and Edmonds present a numerical approach to studying how abandoned channels affect flow routing and channel stacking patterns in alluvial fans. The authors formulate a randomwalk model that finds a route for water and sediment on a fan surface, thereafter evolves the channel bed in one dimension along this path until an avulsion criterion is met, and then re-routes the flow according to a random walk until a new path is forged. This relatively simple algorithm is decorated with a few extra rules, which turn out to make a great deal of difference in the outcomes. The first, and most important rule, is that previous channels in the landscape can either be a preferred path for the random walk, or an unpreferred path for the random walk when the algorithm is in the routefinding phase. This attractive or repulsive quality of the abandoned channels is a continuous variable for each that modifies the probability of a given random walk cell. The second rule is that channels do not persist on the landscape forever, and are "healed" according to one of three procedures: (1) eroding high elevations until only swales remain, (2) filling depressions until only ridges remain, or (3) the topography both raises and lowers until no topographic features remain. Their model design is motivated by observations from large alluvial fans, where the authors see a large density of relict alluvial ridges, as well as a lower density of channels beyond some critical distance from the mountain front. The authors find that with these two continuous variables, and three mechanisms for abandoned channel modification, they can produce a rich diversity of outcomes in the model. In particular, they find that only the third (3) mode of channel healing is capable for achieving a steady-state fan, and that the degree that channels either repulse or attract reoccupation fundamentally shifts where avulsions occur in a strike-average sense. Moreover, their model results broadly mimic the general topographic features of the fans they drew inspiration from, lending some credence to the approach.

- We appreciate Dr. Barefoot's thoughtful review of our manuscript and kind words regarding its novelty. The reviewer provided useful and comprehensive feedback on improving the understandability of figures and tables, and we were able to make all changes except for the colorbar on Figure 2 (for reasons provided below). We thank the reviewer for their efforts and feedback.

- Note: line numbers referenced by the reviewer refer to the original manuscript, while (unless otherwise explicitly stated) line numbers in our responses refer to the revised manuscript with tracked changes accepted.

**Overall Comments**

I found this manuscript to be clear, well-structured, and very detailed. The model design makes a lot of sense, and I think the authors have shown a few very intuitive outcomes while also demonstrating a few less intuitive ones that spark interest. In particular, I thought the outcome where avulsion locations shift basinward when abandoned channels are barriers to flow was very intuitive, and makes for a satisfying result. In contrast, I found it surprising that imposing a rule

that only negative or positive relief can be erased can drive the model to never achieve steady-state. These outcomes are presented and framed well, the conclusions are well-supported and impactful. My constructive comments are limited to a few minor comments on the visual presentation of the figures, and a few clarification questions on a few modeling choices. Other than these, I recommend the article be published. I look forward to citing this paper when my future work involves the stratigraphic architecture of fans.

**Minor Comments**

1. I have a question about this modeling choice. I am not sure if I understand why the simulation has to abort if a timestep results in a failed routing. If this were a real fan, the avulsion does not get a do-over, it has to fill the pond until it overflows, and then carries on its way. I wonder if by imposing this rule, you've introduced an artificial artifact of channel choice, where avulsions from the far-distant past can prohibit the present channel from traversing an entire sector of the fan. What if you adopted a really simple flooding algorithm instead? If while doing the random walk, the river encounters a dead end, it floods the area until it finds the nearest low point, and then starts routing from there. line 325

> - The reviewer raises an excellent point. It's certainly true that there is likely a wide variety of possible ways that a failed avulsion can progress. The reviewer's scenario makes sense as one possible outcome, and we had previously considered ourselves that a "spill-and-fill" model of failed avulsion pathfinding would result in abandoned channel levees no longer being complete repulsors to pathfinding flow. However, it is incorrect to say that on real fans avulsions do not get "do-overs". Failed avulsions are commonplace as crevasses have insufficient immediate slope to "succeed" (Slingerland and Smith 2004). In fact, we view every healed crevasse as a failed avulsion, and these are routinely observed on fans. We ourselves have observed frequent, repeated, and sometimes failed avulsions in the Landsat remote sensing record in a related study, cited in lines 98-99 of this manuscript (Edmonds et al. 2022, at https://doi.org/10.1130/G49318.1). One example would be the attempted avulsion c. 2015-2020 on the Rapulo river, which is excellently viewable via Google Earth Engine's Timelapse tool (link to the exact view at: https://earthengine.google.com/timelapse#v=-14.64137,-66.50818,10.136,latLng&t=2.93&ps=50&bt=19840101&et=20201231&startDwell=0&endDwell=0). This avulsion initiates into a lake adjacent to the channel (flowing towards the mountain for a short span!) then pathfinds through river-right forests before rejoining a downstream abandoned channel. The flow never collapses into a single coherent channel, however, and instead eventually returns to the original pathway and avulses to reoccupy an abandoned channel on river-left. This entire scenario plays out over a shorter timescale than the decadal timestep used in our model. This scenario more closely resembles our "do-over" implementation, though there are undoubtedly some geomorphic effects of such a failed avulsion that we do not include. In sum, the reviewer correctly puts their finger on an important mechanism that is under-understood and that deserves further exploration. We have added lines to our methods (374-379) to explain to the

reader that while our implementation matches the limited remote sensing record, it is undoubtedly simplified compared to the breadth of possible real world outcomes.

2. I think you mean Equation 12 instead of 17? line 341

- Yes we did! Thank you.

**Figures**

– Throughout, I found myself struggling with the choices of colorbar used here. The authors are using parula, I think, which is a marked improvement in Matlab to the previous default colorbar, jet. However, parula is still not perceptually uniform. If a sequence of numbers that was strictly linear was plotted in parula, a viewer would perceive nonlinear jumps in intensity along the gradient. Put another way, there are features in the colormap that show up in plots that are not features of the dataset. To plot elevations, maybe try a single-hue colormap, or winter which I think is perceptually uniform.

- The reviewer is correct that we used parula to plot model output elevations. We chose parula because it is friendly to color vision deficiencies (which we tested using the Coblis – Color Blindness Simulator), and (we thought) perceptually uniform. Others have noted this about parula (ex. Nuñez et al. 2018, at https://doi.org/10.1371/journal.pone.0199239) and Hughes 2019, at https://brushingupscience.com/2019/10/01/default-colormaps-are-parula-and-viridis-really-an-improvement-over-jet/). Moreover, Matlab described parula as uniform in their original justification posts for switching from Jet to Parula (https://blogs.mathworks.com/steve/2014/10/20/a-new-colormap-for-matlab-part-2-troubles-with-rainbows/). In particular, the middle link to Hughes' blog post has this small section worth reproducing:

  The primary criterion in developing Parula and Viridis was to ensure the default colourmaps are perceptually uniform. One way to interpret this is that it means that if the colourmap is converted to grayscale, it should be linear. Parula and Viridis certainly achieve that, albeit with a limited range between light and dark for Parula.

[Figure]

  With that said, it appears on further review of an early matplotlib colorbar blog post, that while parula may be nearly perceptually uniform, it is not perfectly so! (https://bids.github.io/colormap/). As such, we imported a Viridis colorbar to Matlab and re-ran models to update the colorbar for all planform figures. These outputs have been replaced.

– In general, I have one piece of feedback that applies to all your figures. As a point of style, you seem to have opted to putting a heavy black frame around every plot. While in the design phase, I can imagine it is helpful to have such a frame to see spacing between elements. For a finished

product though, it intrudes on the visual space and commands attention, subconciously distracting the reader from the contents of the figure. Rules, used judiciously, can establish visual hierarchy (Figure 1 is a good example), but in a lot of cases here it is just too much. For all of your figures, I recommend getting rid of the bounding boxes.

- Thank you for the suggestion. You're right that the black bounding boxes helped with layout and framing, and were present so that the figures could be reproduced at the specific size and shape desired on paper without any rescaling. As such, at this stage in production, they are no longer necessary. We have removed them (where present) from Figures 2, 4, and 6-12. I kept them in Figures 1, 3, and 5 in order to preserve necessary visual hierarchy.

– For example, here in Figure 2, the boxes around the annotations are essential, because otherwise the reader will never see them. However, I would remove the box around the figure, and take away the boxes around each of your colorbars. For these elements, proximity is all you need to establish a connection.

- Fixed.

- On the subject of color, I would recommend a different colormap [for Figure 2]. The one here is distorting the visual presentation of the data. This colormap is really good at highlighting contrast in certain parts of its spectrum (e.g. yellow-to-red), and so the nice contrast showing the ridges that you want to see is limited to an arcuate band halfway through each map. I might instead recommend making four maps. In one pair, show just elevation in a single-hue colormap from light to dark so that the reader can see the conical shape. In the other pair, compute the slope map and plot that in a different single-hue colorbar. That way we can see both the ridges and the overall shape, but separated into two panels.

- We agree that the colorbars in Figure 2 are non-perceptually linear. We tried to implement the reviewer's suggestions by plotting both elevation maps and slope maps with linear monochromatic colorbars for both megafans; see results below for the Rapulo (Figure 2A), where lighter colors represent higher elevations and slopes.

[Figure]

We believe that our original colorbar, while not perfect, does a better job of communicating the abandoned channel topographic features while also communicating the overall slope of the megafans. Unfortunately, this is a particularly challenging geomorphic feature to show using the data provided, because of at least two reasons. First, while megafans are generally low relief features, there still remains a much greater drop in elevation between the highest and lowest values shown in the figure (~70 m

change) relative to the elevation of abandoned channels (~meters scale); as the reviewer foreshadowed, the signal of downslope change makes it hard to observe abandoned channels. Normally this would be a good opportunity to use a slope map to show local features, as the reviewer suggested, but unfortunately, results in this case are poor. This is because the features are small relative to the noise present in the radar-derived and corrected SRTM (BEST) used in our analysis. This setting is just too low-relief to get quantitatively meaningful results from our imagery. For the same reason, hillshades, semi-transparent slope maps, and other analyses were tried but did not yield a product that was more effective at communicating the degree of floodplain channelization than the original colormap.

- With that said, we do agree with the importance of making readers aware of some of the pitfalls of non-perceptually uniform colormaps, and have therefore updated the figure caption (161-162) to bring this matter to the reader's attention.

– Figure 3 is very nice, and seems intuitive and helpful, but it appears to have lost its caption.

- You are correct, thanks for pointing this out.,

– I like Figure 4 a lot. It's very helpful.

- Thank you!

– In Figure 7, why do you think there is this odd, smoothly-sinusoidal lobe-switching? I didn't see it discussed in detail, but this is shockingly regular, and only seems to occur in the deposition-only or erosion-only healing modes

- We agree that this was a result in need of more exploration in our manuscript, and as such, have added content to sections 5.5 (614-637) and 6.3 (690-696).

**Tables**

Usually for a manuscript, table design is not super important, but for ESurfD, it seems that they simply publish tables as-is, instead of reformatting them. Since this is the case, I have a few constructive comments that will make your tables much more legible.

– Vertical rules are not a great way of guiding a reader's eye. Alignment is a much better tool in the vertical direction […] Horizontal rules are great for breaking up your tables visually into topical or related sections and for connecting items across rows, which is much harder for human eyes on the written page.

- While we don't disagree, our reading of the ESurf author guidelines (at https://www.earth-surface-dynamics.net/submission.html#figurestables) suggests that "Horizontal lines should normally only appear above and below the table, and as a separator between the head and the main body of the table." We took a quick glance at some recent papers, however, and found examples of tables both with (ex. https://doi.org/10.5194/esurf-10-97-2022) and without (ex. https://doi.org/10.5194/esurf-

) horizontal rules throughout tables, so I'm not sure how firm this particular guideline is.

– So for Table 1, remove the box around the outside, and leave just the horizontal rule separating the heading from the table elements. See if you can make the individual cells single-spaced, while leaving some breathing room between cells. All the same for Table 2.

- Done as suggested. We don't feel too strongly about this, so we are happy to hear from the editor/production if they prefer the old or new formatting for the table.

– For Table 3, do the same things like tightening the spacing within cells while leaving space between cells, also consider making the headings for each column bold. I would put the first column for figure references at the end. Also, the bit about having a parameter marked as "variable, see figure X" is very confusing, and forces your reader to flip back and forth. Instead, I would have groups of rows (set apart with horizontal rules) where you show every model run with every parameter combination. I know its a lot, but this table is already almost a page, so why not just make it a well-designed full-page table and go for it?

- As requested, we put the figure title column at the end, added the range of parameter values to the table (instead of referring to the tables), removed all vertical rules, and used horizontal rules to separate runs. We also bolded the headers and did the same to Tables 1 and 2.